# Irreversible electroporation reverses resistance to immune checkpoint blockade in pancreatic cancer

Jun Zhao[1], Xiaofei Wen[1,2,3], Li Tian[4], Tingting Li[1,5], Chunyu Xu[6], Xiaoxia Wen[1], Marites P. Melancon[4], Sanjay Gupta[4], Baozhong Shen[2,3], Weiyi Peng [6] & Chun Li [1]

Immunotherapy has only limited efficacy against pancreatic ductal adenocarcinoma (PDAC) due to the presence of an immunosuppressive tumor-associated stroma. Here, we demonstrate an effective modulation of that stroma by irreversible electroporation (IRE), a local ablation technique that has received regulatory approval in the United States. IRE induces immunogenic cell death, activates dendritic cells, and alleviates stroma-induced immunosuppression without depleting tumor-restraining collagen. The combination of IRE and anti-programmed cell death protein 1 (anti-PD1) immune checkpoint blockade promotes selective tumor infiltration by CD8$^+$ T cells and significantly prolongs survival in a murine orthotopic PDAC model with a long-term memory immune response. Our results suggest that IRE is a promising approach to potentiate the efficacy of immune checkpoint blockade in PDAC.

[1] Department of Cancer Systems Imaging, The University of Texas MD Anderson Cancer Center, Houston, TX 77054, USA. [2] Molecular Imaging Research Center of Harbin Medical University, Harbin 150001 Heilongjiang, China. [3] Department of Interventional Radiology, Fourth Hospital of Harbin Medical University, Harbin 150001 Heilongjiang, China. [4] Department of Interventional Radiology, The University of Texas MD Anderson Cancer Center, Houston, TX 77030, USA. [5] Department of Biophysics, School of Life Science & Technology, University of Electronic Science and Technology of China, Chengdu 610054 Sichuan, China. [6] Department of Melanoma Medical Oncology–Research, The University of Texas MD Anderson Cancer Center, Houston, TX 77054, USA. These authors contributed equally: Jun Zhao, Xiaofei Wen. Correspondence and requests for materials should be addressed to C.L. (email: cli@mdanderson.org)

I mmune checkpoint blockade is showing promise in cancer treatment and producing durable responses in several tumor types[1]. Its efficacy in treating patients with pancreatic ductal adenocarcinoma (PDAC), however, is limited by the immunosuppressive stroma associated with this cancer[2]. PDAC is characterized by a highly fibrotic stroma that can physically exclude cytotoxic T cells from the vicinity of tumor cells. The immunosuppressive microenvironment within the stroma can also dampen the activity of infiltrating T cells[3,4].

Recent attempts to modulate PDAC stroma have generated mixed results. Genetic depletion of fibroblast activation protein alpha-positive (FAPα+) cancer-associated fibroblasts (CAFs) improved the efficacy of anti-PDL1 blockade[5]. Inhibition of focal adhesion kinase-1 relieved stromal fibrosis, decreased infiltration of immunosuppressive cells, and subsequently enhanced the efficacy of anti-PDL1 therapy[6]. In contrast, depletion of the alpha smooth muscle actin-positive (αSMA+) CAFs led to the loss of collagenous matrix, promoted infiltration by immunosuppressive T regulatory cells (Tregs), and produced an alarmingly aggressive phenotype of PDAC[7,8]. Further studies suggested that stromal elements can restrain PDAC from an unchecked growth[9]. On the other hand, systemic injection of stroma-modulating agents can cause adverse effects in healthy organs. For example, PEGylated recombinant human hyaluronidase, although it successfully increased tumor perfusion by degrading hyaluronic acid in PDAC stroma, caused significant musculoskeletal toxic effects in a clinical trial (NCT0083470)[10]. Taken together, these results indicate the potential therapeutic benefit of modulating the stroma via a local approach while preserving the tumor-restraining collagenous matrix of PDAC.

Irreversible electroporation (IRE) is a novel interventional technique for the local ablation of PDAC; it has been approved for clinical use in the US by the Food and Drug Administration[11,12]. Although reversible electroporation has been used for decades for delivery of genes and drugs into tumor cells[13], the use of IRE for tumor ablation was introduced only recently by Davalos et al.[14]. IRE uses short high-voltage electric pulses to induce cell death through permanent membrane lysis or loss of homeostasis[15–17]. In addition to killing tumor cells, IRE also increased the delivery of gemcitabine to PDAC tumor[18], suggesting a modulation of the PDAC stroma; but the exact extent of stromal change remains unclear. Meanwhile, recent studies on other tumor models, including a rat sarcoma[19], a murine renal carcinoma[20], and a canine glioma model[21], have shown an improved antitumor efficacy of IRE in immunocompetent animals, indicating a possible role of the host immune system. However, these studies were not performed in the context of immunotherapy. Neither did these studies investigate stromal modulation. Up to date, it is unknown whether IRE can potentiate the antitumor efficacy of immunotherapy in the poorly immunogenic PDAC.

Based on these analyses, we hypothesized that IRE enhances the efficacy of anti-PD1 therapy in PDAC by activating the immune system and alleviating stroma-induced immunosuppression. The preclinical results reported here demonstrate that the combination of IRE and anti-PD1 promoted tumor infiltration by CD8+ cytotoxic T cells without recruiting other immunosuppressive cells, and significantly prolonged survival in an orthotopic murine PDAC model. Importantly, the IRE + anti-PD1 treatment achieved a cure rate of 36–43% with a memory T cell response. Our findings suggest that the combination of IRE with immune checkpoint blockade as a promising and safe strategy for treating patients with PDAC is warranted.

## Results

**IRE enhanced PD1 blockade in pancreatic cancer and melanoma.** We first evaluated the antitumor efficacy of IRE and anti-PD1 immune checkpoint blockade in a murine orthotopic PDAC model (KRAS* model) with an inducible mutation in *Kras* (*Kras^{G12D}*), an oncogenic driver mutation for PDAC. The experimental set up and treatment schedules are illustrated in Fig. 1a. The median survival was 6 days after enrollment for untreated control mice, 8 days for anti-PD1-treated mice, and 11.5 days for IRE-treated mice. In comparison, the median survival of mice treated with the combined IRE + anti-PD1 was 31.5 days, significantly longer than that of any of the other groups ($p < 0.0001$, log-rank test). Four of the 11 (36%) mice treated with IRE + anti-PD1 were alive at the end of the 60-day study period with no palpable tumor (Fig. 1b). To confirm this finding, we repeated the IRE + anti-PD1 treatment in additional 7 KRAS*-bearing mice: 3 of these 7 mice (43%) reached the 60-day study endpoint with no palpable tumor (Supplementary Figure 1). Histopathological analysis of major organs showed that IRE + anti-PD1 had minimal impacts on these organs (Supplementary Figure 2).

We extended our findings to the murine B16F10 melanoma model that, once established, is resistant to anti-PD1 therapy[22]. The median survival was 5 days after enrollment for untreated controls, 5.5 days for mice that received anti-PD1 only, and 8 days for those that underwent IRE only (Fig. 1c). The median survival of IRE + anti-PD1-treated mice was 23 days, significantly longer than that of any of the other groups; while two mice reached the 60-day survival endpoint. These data indicate that IRE + anti-PD1 could generate durable antitumor responses in solid tumors that are resistant to anti-PD1 therapy.

We then used axial $T_2$-weighted magnetic resonance imaging ($T_2$-MRI) to monitor the growth of KRAS* tumors in a separate small-scale study. Representative MRI images are shown in Fig. 1d; the whole imaging set is included in Supplementary Figures 3–6. The control mice exhibited an aggressive tumor growth, and had to be euthanized due to excessive tumor burden on day 9. Mice in the monotherapy groups experienced a slower tumor progression than the sham control. Still, they had to be euthanized on day 16 due to excessive tumor burden. In contrast, two out of the five mice (40%) receiving IRE + anti-PD1 showed no signs of tumor on MRI images by day 42. These imaging results were consistent with the survival data shown in Fig. 1b.

**Adding anti-CTLA4 did not further prolong animal survival.** Concomitant treatment with antibody to cytotoxic T lymphocyte-associated antigen-4 (anti-CTLA4) and anti-PD1 was previously reported to generate a rapid and strong tumor regression in melanoma patients[23]. We investigated whether adding anti-CTLA4 would further prolong survival in the KRAS* model. IRE + anti-CTLA4 + anti-PD1 resulted in a median survival of 41 days, significantly longer than those of the sham control, IRE only, or anti-PD1 + anti-CTLA4 groups (Fig. 2a, $p < 0.0001$, log-rank test). Five of the 11 mice (45%) in the IRE + anti-CTLA4 + anti-PD1 group were alive at the 60-day study endpoint, although the median survival of this group was not significantly different from that of the IRE + anti-PD1 group (Fig. 2b). Notably, mice treated with IRE + anti-CTLA4 + anti-PD1 experienced more body-weight loss than those treated with IRE + anti-PD1 (Fig. 2c) at 4 days after the initiation of treatment, suggesting a substantial toxicity. As a result, we focused on the combination of IRE + anti-PD1 in further studies.

**Anti-tumor efficacy was due to infiltrating CD8+ T cells.** We then profiled immune cells in KRAS* tumor at 9 days after IRE

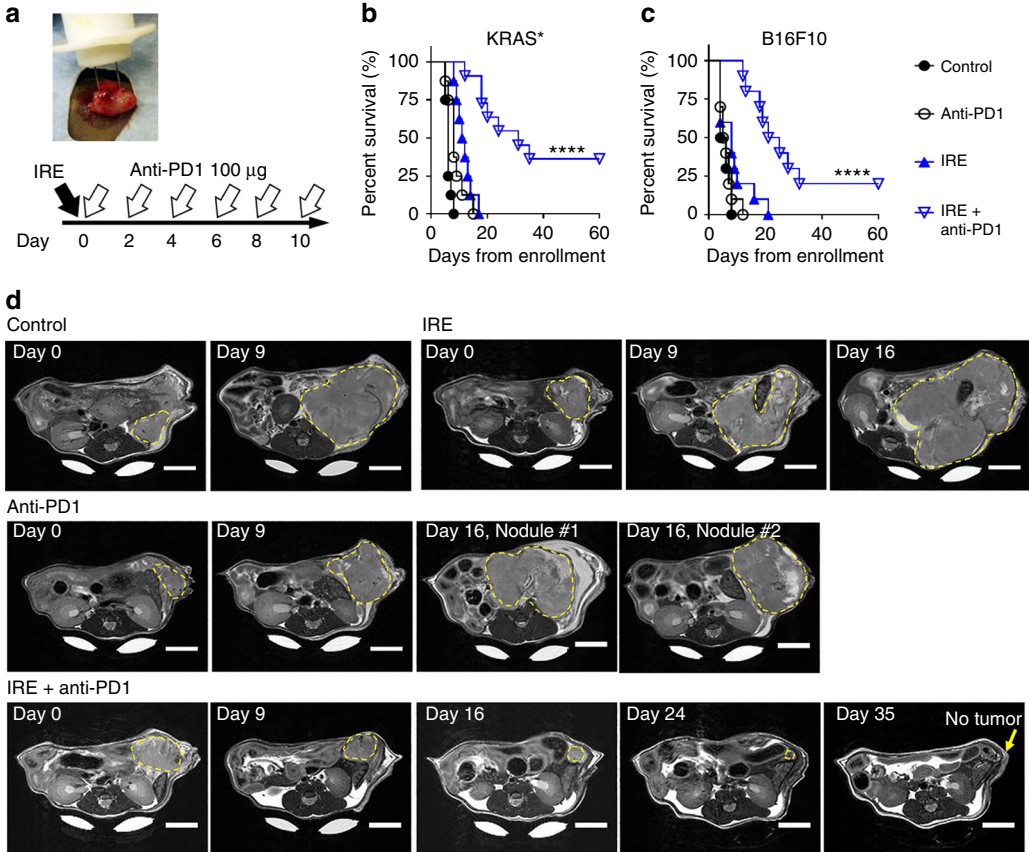

**Fig. 1** Animal survival after treatment with IRE and/or anti-PD1. **a** Treatment schedule and a photograph showing the placement of the two-electrode IRE array in the KRAS* tumor. C57BL/6 mice bearing orthotopic KRAS* tumors were enrolled for treatment once tumor size reached about 7 mm in one dimension. Sham surgery was performed on both control and anti-PD1 treatment groups. **b** Kaplan–Meier survival analysis of mice with KRAS* tumor treated with sham control (black solid circle, $n = 8$), IRE (black circle, $n = 8$), anti-PD1 (blue solid triangle, $n = 8$), or IRE + anti-PD1 (blue triangle, $n = 11$). ****$p < 0.0001$, log-rank test. **c** Survival analysis of B16F10-bearing mice treated with sham control ($n = 10$), IRE ($n = 10$), anti-PD1 ($n = 10$), or IRE + anti-PD1 ($n = 10$). ****$p < 0.0001$, log-rank test. **d** Representative axial $T_2$-MRI images of a KRAS*-bearing mouse from each group in a separate small-scale study. Three mice per group were enrolled in the sham control, anti-PD1, and IRE groups; 5 mice were enrolled in the IRE + anti-PD1 group. Treatment started on day 0. MRI slices with the largest tumor cross-section are presented to show tumor size at each time point. The entire extent of the tumor is outlined with yellow dashed lines. MRI images were acquired weekly until the mice were euthanized due to excessive tumor burden. The surviving mice in the IRE + anti-PD1 group were imaged until day 42 (Supplementary Figure 6). Scale bar = 5 mm. Source data are provided as a Source Data file

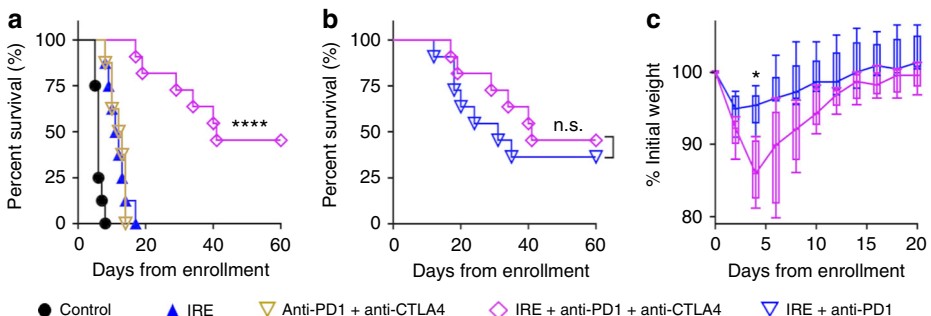

**Fig. 2** Survival of KRAS* model treated with IRE, anti-CTLA4, and/or anti-PD1. **a** Kaplan–Meier survival curves of mice in sham control (black solid circle, $n = 8$), anti-CTLA4 + anti-PD1 (brown triangle, $n = 8$), IRE only (blue solid triangle, $n = 8$), and IRE + anti-CTLA4 + anti-PD1 groups (magenta diamond, $n = 11$). ****$p < 0.0001$, log-rank test. **b** Survival of IRE + anti-CTLA4 + anti-PD1 ($n = 11$) and IRE + anti-PD1 (blue triangle, $n = 11$) groups, n.s. not significant, log-rank test. **c** Relative change in mouse body weight after treatment with IRE + anti-CTLA4 + anti-PD1 or IRE + anti-PD1 ($n = 5$ each group). Data presented as interquartile range (IQR) with a median center line and min to max error bars, *$p < 0.05$, double-sided Student's $t$-test. Source data are provided as a Source Data file

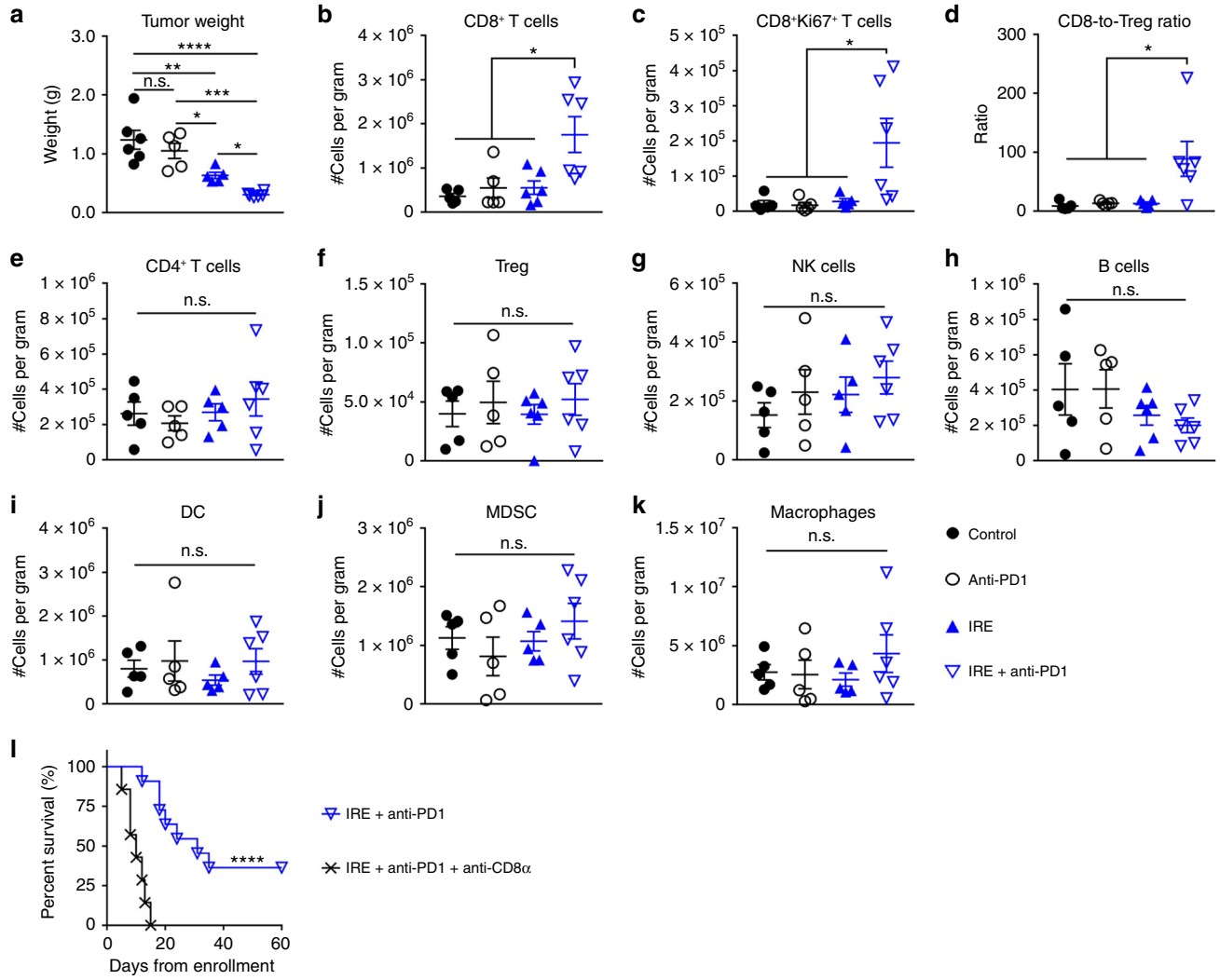

**Fig. 3** Profiling of intratumoral immune cells in KRAS* tumors. **a** Tumor weights at 9 days after initiation of treatments. **b** Frequency of CD8$^+$ T cells (CD3$^+$CD8$^+$). **c** Frequency of proliferating CD8$^+$ T cells (CD8$^+$Ki67$^+$). **d** CD8$^+$-to-Treg cell ratio. **e** Frequency of CD4$^+$ T cells (CD3$^+$CD4$^+$). **f** Frequency of Tregs (CD4$^+$CD25$^+$Foxp3$^+$). **g** Frequency of NK cells (NK1.1$^+$). **h** Frequency of B cells (CD19$^+$). **i** Frequency of DCs (CD11c$^+$CD11b$^-$). **j** Frequency of MDSCs (CD11c$^+$CD11b$^+$Ly6C$^+$ and CD11c$^+$CD11b$^+$Ly6G$^+$). **k** Frequency of macrophages (F4/80$^+$). Five tumors per group from the sham control (black solid circle), anti-PD1 (black circle), and IRE (blue solid triangle) groups, and 6 tumors from IRE + anti-PD1 group (blue triangle) were collected 9 days after initiation of treatments. Data are presented as mean ± standard error of mean (SEM). Significance of differences was determined using 1-way ANOVA followed by Tukey post hoc analysis. **l** Kaplan–Meier survival curves showing the effect of CD8 neutralization. Mice in the IRE + anti-PD1 + anti-CD8α (black x, $n = 7$) had significantly shorter survival compared to that in the IRE + anti-PD1 group ($n = 11$), log-rank test. *$p < 0.05$, **$p < 0.01$, ***$p < 0.001$, ****$p < 0.0001$, n.s. not significant. Source data are provided as a Source Data file

treatment, at which time 5 doses of anti-PD1 had been injected for the anti-PD1 and combination groups. The mean tumor weight was greatest in the untreated controls (1.24 ± 0.16 g); it was 1.05 ± 0.13 g in the anti-PD1 only group, 0.63 ± 0.05 g in the IRE only group, and 0.30 ± 0.02 g in the IRE + anti-PD1 group (mean ± SEM, $n = 5$ or 6, Fig. 3a). Tumors in the IRE + anti-PD1 group were significantly smaller than those of others. The numbers of representative immune cells per gram of tumor tissue are presented in Fig. 3b–k, and the corresponding representative flow cytometry plots and gating strategies are shown in Supplementary Figures 7–9. There were significantly higher frequencies of total CD8$^+$ T cells and Ki67$^+$CD8$^+$ proliferating T cells, as well as a higher CD8-to-Treg ratio in the IRE + anti-PD1 group than in any of the other groups (Fig. 3b–d). There were no significant differences in frequencies of CD4$^+$ T cells, Tregs, natural killer (NK) cells, B

cells, DCs, myeloid-derived suppressive cells (MDSCs), or macrophages between the IRE + anti-PD1 group and the other treatment groups (Fig. 3e–k). These results suggested that IRE + anti-PD1 not only selectively increased the infiltration by CD8$^+$ T cells, but also enhanced their proliferation.

Our results suggested that the activated CD8$^+$ T cells were associated with the superior anti-tumor efficacy of the IRE + anti-PD1 combination. We then performed a depletion study to deplete CD8$^+$ T cells with a CD8-neutralizing antibody (anti-CD8α) (Fig. 3l). The median survival of the IRE + anti-PD1 + anti-CD8α group was 10 days, significantly shorter than that of the IRE + anti-PD1 group (31 days, $p < 0.0001$, log-rank test). There was no long-term survivor in the CD8$^+$ T cell depleted treatment group. Taken together, our data support that activated CD8$^+$ T cells played a central role in IRE-mediated sensitization of anti-PD1 therapy.

**IRE and anti-PD1 induced long-term memory T cells**. No palpable tumors were present among the KRAS*-bearing mice that had received the IRE + anti-PD1 treatment and survived for 60 days. To investigate whether there was a memory antitumor response, these long-term surviving mice ($n = 10$) were re-challenged with KRAS* cells by subcutaneous inoculation. Age-matched healthy mice that went through sham surgery at the same time of the treated mice were used as controls. The control mice exhibited a robust tumor growth (Fig. 4a). In contrast, all KRAS* mice that lived 60 days after IRE + anti-PD1 treatment rejected the subcutaneous tumor challenge and were tumor-free throughout the duration of the study. To further analyze the systemic T cell response, 6 out of the 10 re-challenged long-term surviving mice were euthanized at 9 weeks after the tumor cell re-challenge and their splenocytes collected. These splenocytes were incubated with KRAS* tumor lysate in the presence of murine bone marrow-derived DCs. The long-term surviving mice had 2.3 times more interferon gamma (IFN-γ)-secreting splenocytes than the treatment-naive mice, as quantified by the ELISPOT assay (Fig. 4b). Compared to splenocytes from the control mice that had not been exposed to KRAS* cells, the splenocytes from the long-term surviving mice comprised 2.1-fold higher frequency of CD4+ memory T cells (CD4+CD44+CD62L−) and 5.7-fold higher frequency of CD8+ memory T cells (CD8+CD44+CD62L−) (Fig. 4c)[24]. Of the remaining 4 mice, all of them survived for more than 9 months from the initiation of treatment (7 months after tumor cell re-challenge). Histological examination revealed no microscopic tumor nodules in their pancreas (Fig. 4d, Supplementary Figure 10).

**IRE induced release of danger-associated molecular patterns**. To investigate the mechanisms underlying the durable responses by IRE + anti-PD1, we first evaluated in vitro the mode of cell death induced by electroporation in a 4-mm gap cuvette. A pilot study indicated that electric pulses at a voltage of up to 200 V did not have a significant impact on the viability of KRAS* cells (Supplementary Figure 11). Therefore, 200 and 960 V were selected for further experiments. Neither KRAS* nor B16F10 cells showed substantial changes in percentages of apoptotic or necrotic cells at low voltage (200 V, Fig. 5a). In contrast, more than 98% of cells that underwent IRE at high voltage (960 V) were double-stained for Annexin V and propidium iodide (PI) within 30 min, indicating a rapid induction of late apoptotic/necrotic cell death. The supernatants of IRE-treated cells were analyzed for adenosine triphosphate (ATP) and high-mobility group protein B1 (HMGB1), two validated DAMPs. Pulses at 200 V caused no change in concentrations of either ATP or HMGB1 in the KRAS* cells and only moderate increases in ATP concentration in the B16F10 cells. In contrast, IRE at 960 V increased the extracellular ATP concentration by 11 times in KRAS* cells and 7.7 times in B16F10 cells, and the extracellular HMGB1 concentration by 12.7 times in KRAS* cells and 8.6 times in B16F10 cells compared to untreated cells (Fig. 5b). To investigate whether IRE-treated KRAS* cells mediated activation and maturation of DCs, live KRAS* cells or IRE-treated KRAS* cells were incubated with bone marrow-derived murine dendritic cells (DCs) for 24 h. Flow cytometry analyses (Fig. 5c) showed that IRE-treated KRAS* cells increased the expression of DC activation/maturation markers CD40, CCR7, and CD86 by $72 \pm 2\%$, $52 \pm 4\%$, and $51 \pm 4\%$ (mean ± SEM, $n = 3$), respectively, compared to live KRAS* cells, in the tumor cell-exposed DCs ($p < 0.01$, two-sided Student's t-test).

**IRE transiently modulated stroma in favor of immunotherapy**. Given that the combination of IRE + anti-PD-1 was able to increase the intratumoral frequency of CD8+ T cells, we next studied to what extent IRE modulated the stroma of KRAS* tumors; because it is known that modulation of PDAC stroma enhances tumor infiltration of CD8+ cells[25]. IRE induced substantial necrosis in the tumor centers that was surrounded by a rim of viable tumor (Fig. 6a). Immunohistochemical (IHC) staining in the viable region revealed that, at 4 days after treatment, IRE induced a transient increase in microvessel density (MVD) in the viable tumor region ($7.1 \pm 0.5\%$ CD31+ pixels per 200× visual field, mean ± SEM, $n = 10$–15) compared to untreated controls ($2.0 \pm 0.3\%$, Fig. 6b). MVD fell to $3.3 \pm 0.3\%$ at 6 days after IRE. On the other hand, the expression of FAPα at 4 days after IRE ($4.0 \pm 0.9\%$ FAPα+ pixels per 200× visual field) was half that of control ($8.2 \pm 0.7\%$), which then return to $7.9 \pm 1.3\%$ at 6 days after IRE (Fig. 6c).

At the protein level, hypoxia-inducible factor 1-alpha (HIF-1α) expression in the viable region of tumors at 4 days after IRE (Fig. 6d) was 53% that in controls. The expression level of the hypoxia marker carbonic anhydrase 9 (CA-IX) was 24% that in controls, hyaluronic acid binding protein 1 (HABP1, a marker for the matrix hyaluronic acid) was 70% that in controls, lysyl oxidase (LOX, a marker for the rigidity of the extracellular matrix) was 41% that in controls, and PD-L1 was 18% that in controls. The expression of these proteins mostly rebounded back at 6 days after IRE, with the exception of HABP1, which remained at 74% that in untreated controls. There was no significant change in the expression of αSMA at either 4-day or 6-day time point after IRE.

FITC-conjugated dextran was used to determine whether IRE increased the permeability of tumor blood vessels (Fig. 6e, f)[26]. The mean fluorescence intensity (MFI) of FITC in the tumor sections at 4 days after IRE was 18.7 times higher than that of untreated tumors; this value dropped to 8.4 times that of control at 6 days after IRE (Fig. 6e). Co-immunofluorescence staining of blood vessels with CD31 (Fig. 6f) showed that FITC-dextran extravasated into the interstitial space of KRAS* tumor at 4 days after IRE. Taken together, these findings indicate that IRE transiently modulated tumor stroma by increasing MVD and tumor blood vessel permeability, softening extracellular matrix (as indicated by reduced LOX and depletion of hyaluronic acid), and alleviating hypoxia, all effects that favor tumor infiltration by cytotoxic T lymphocytes[5,27–29].

**IRE + anti-PD1 induced a sustained modulation of stroma**. While tumor MVD had decreased from the peak value of 7.1% at day 4 post-IRE to 3.3% by post-IRE day 6 (Fig. 6b), it had returned to the baseline level of <2.0% by 9 days post-IRE (Fig. 7a). IRE + anti-PD1 sustained the effect of IRE-alone on MVD. At 9 days after initiation of treatments, at which time 5 doses of anti-PD1 had been administered, the mean MVD of tumors treated with IRE + anti-PD1 was 4 times higher than that from any other group (Fig. 7a). Similarly, the proportion of FAPα+ CAFs in the IRE + anti-PD1-treated group was significantly lower than that in any other treatment group (Fig. 7d), even though the expression level of FAPα+ had returned to the baseline level by 6–9 days after IRE-alone treatment (Figs. 6c and 7d). HABP1 level was significantly lowered by combined IRE + anti-PD1 and IRE alone compared to no treatment (Fig. 7e). The combination of IRE + anti-PD1 significantly suppressed tumor proliferation, showing fewer Ki67+ proliferating cells than any other groups (Fig. 7f). There were no differences between the 6-day and 9-day post-IRE tumors and untreated tumors with regard to expression of type I collagen and αSMA (Fig. 7b, c). These data indicate that anti-PD-1 sustained the impact of IRE on the PDAC stroma for a longer period of time without depletion of type I collagen.

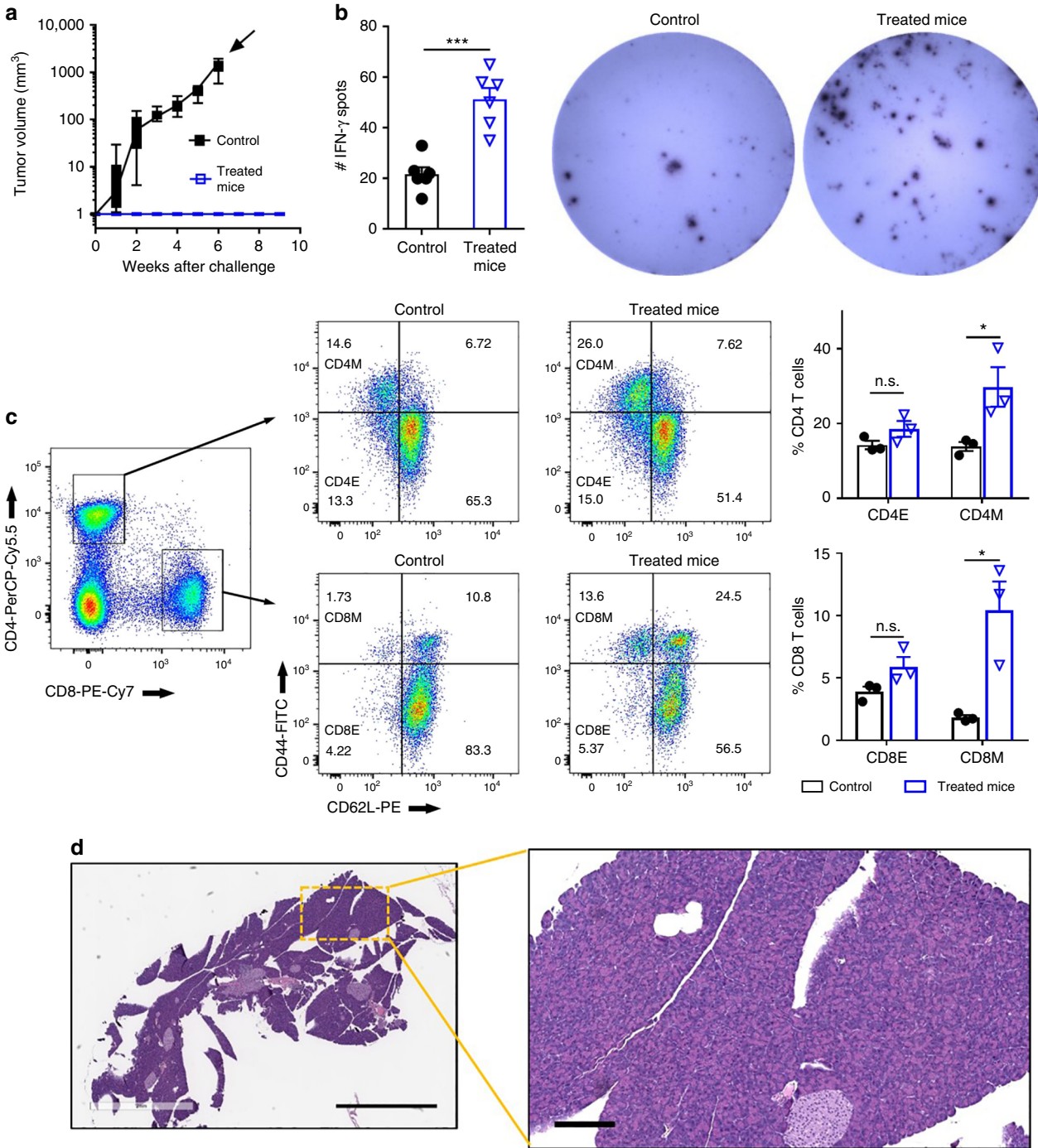

**Fig. 4** IRE + anti-PD1 generated long-term immune responses. **a** Growth curve of rechallenging tumors in the long-term surviving mice treated with IRE + anti-PD1 (blue square, $n = 10$). Age-matched healthy mice that went through sham surgery at the same time of the treated mice were used as controls (black solid square, $n = 9$). Time point of euthanasia is indicated by a solid arrow. Data presented as interquartile range (IQR) with a median center line and min to max error bars. There was no growth of rechallenging tumor in the long-term surviving mice. Their volume was assigned to be 1 mm$^3$ in order to be visible on the semi-logarithmic scale. **b**, **c** Analyses of splenocytes isolated from re-challenged mice (blue triangle) at 9 weeks after inoculation of secondary tumor. Age-matched healthy mice (black solid circle) that went through sham surgery at the same time of the treated mice were used as controls. **b** Quantification of IFN-γ spots and representative images from the ELISPOT assay. Murine bone marrow-derived DCs were pulsed with KRAS* cell lysate, then co-incubated with splenocytes to induce secretion of IFN-γ. Spleens from 3 mice per group were analyzed with a technical duplicate for each spleen ($n = 6$). **c** Representative flow cytometry plots and quantification of effector cells (CD44$^−$CD62L$^−$) and memory cells (CD44$^+$CD62L$^−$) among CD4$^+$ and CD8$^+$ T cells isolated from splenocytes. Three spleens were analyzed for each group ($n = 3$). Data are presented as mean ± SEM in panels (**b**) and (**c**). *$p < 0.05$, ***$p < 0.001$. CD4M, CD4 memory cells (CD4$^+$CD44$^+$CD62L$^−$); CD4E, CD4 effector cells (CD4$^+$CD44$^−$CD62L$^−$); CD8M, CD8 memory cells (CD8$^+$CD44$^+$CD62L$^−$); CD8E, CD8 effector cells (CD8$^+$CD44$^−$CD62L$^−$); n.s. not significant. Significance of difference was determined by two-sided Student's $t$-test. **d** Representative hematoxylin–eosin staining of tumor-free pancreas in KRAS*-bearing mice that were treated with IRE + anti-PD1, and had survived more than 9 months after the initiation of treatments. Scale bar = 2 mm for the overview and 200 μm for the inset. Source data are provided as a Source Data file

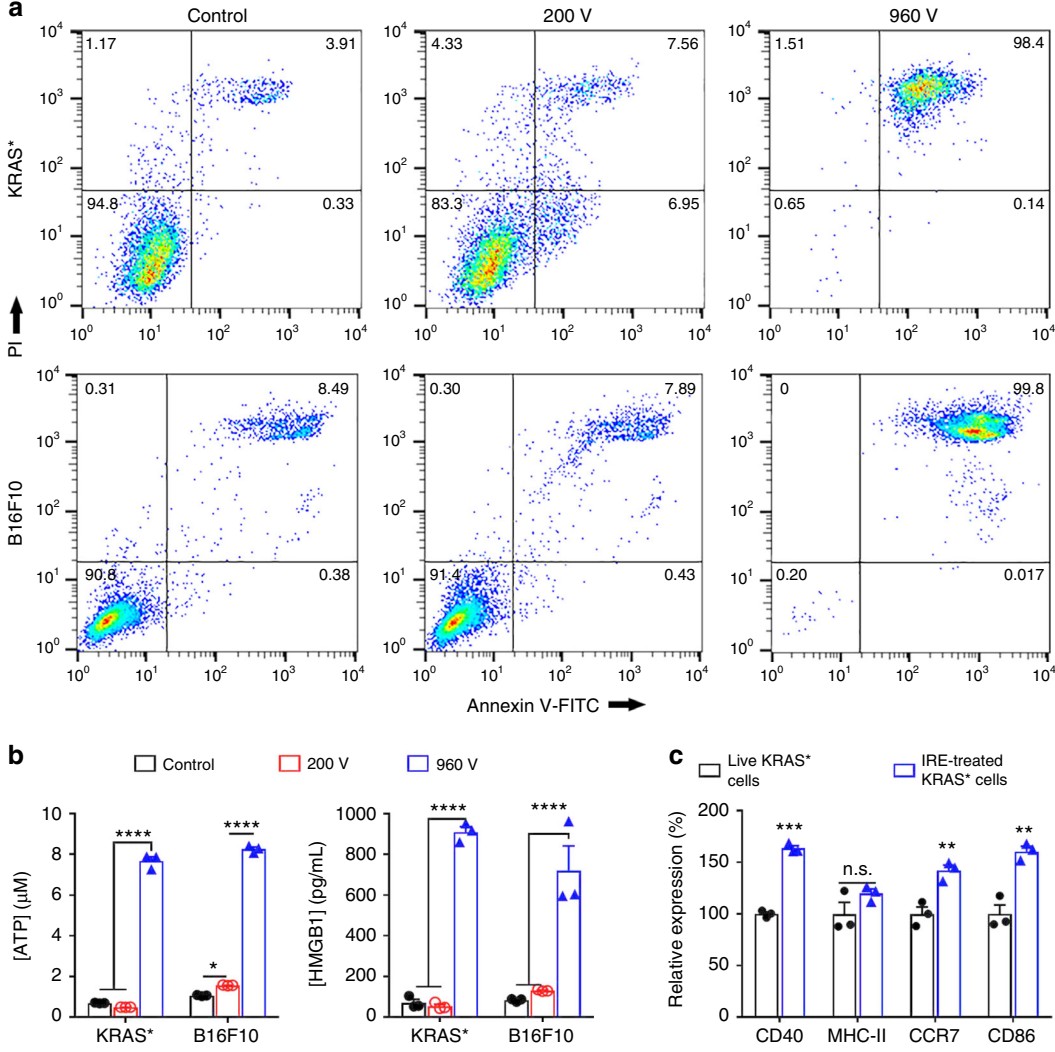

**Fig. 5** In vitro analyses of IRE-treated cancer cells and their impacts on dendritic cells. **a** Annexin V-FITC/PI staining of KRAS* and B16F10 cells. Cells were suspended in phosphate buffered saline (PBS) and electroporated in a cuvette with a 4-mm gap. The parameters for electroporation were: voltage = 200 or 960 V, pulse duration = 100 µs, pulse repetition frequency = 1 Hz, number of pulses = 20. Cells were stained and analyzed within 30 min of treatment. **b** ATP and HMGB1 concentrations in cell supernatants of untreated control (black square), 200 V (red square), and 960 V (blue square) groups (n = 3). **c** Relative expression of DC activation markers, including CD40, MHC-II, CCR7, and CD86, on bone marrow-derived DCs (CD11c+) after incubation with live or IRE (960 V)-treated KRAS* cells for 24 h. Expression was quantified via geometry mean fluorescence intensity (MFI; n = 3). Data are representative of 3 independent experiments and presented as mean ± standard error of mean (SEM). Significance was determined using two-sided Student's t-test. ****p < 0.0001, n.s. not significant. Source data are provided as a Source Data file

**Radiotherapy + anti-PD1 did not produce a durable response.** We next compared IRE with radiotherapy in their ability to enhance immune checkpoint blockade in KRAS* model. Radiotherapy and chemotherapy are first-line treatments for PDAC, and both of them could improve tumor response in combination with immunotherapies[30,31]. We performed an in vitro treatment of KRAS* cells with 10 µM gemcitabine, a first-line medicine for PDAC. Significant cell apoptosis/necrosis was not observed until after 48 h of incubation (Supplementary Figure 12), indicating that chemotherapy-induced death of KRAS* cells was much slower than that by IRE, which occurred within 30 min of treatment. In addition, systemic chemotherapy may impair the host immune system, and as a result may not constitute a fair comparison with the local IRE treatment. Therefore, we focused on radiotherapy as a standard local therapy for the purpose of comparison[32].

We first studied in vitro the mode of KRAS* cell death at 30 min after 10 Gy of radiation and IRE using Annexin V-PI staining (Fig. 8a). Compared to untreated cells, radiation-treated cells had a slight increase in early apoptotic and necrotic cells, and a moderate increase in the Annexin V/PI-double positive cells. In comparison, more than 95% of IRE-treated cells (960 V) were Annexin V/PI-double positive. Analyses of the supernatants of treated cells revealed that 10 Gy of radiation did not cause the release of ATP or HMGB1 at the 30-min time point (Fig. 8b, c). On the other hand, IRE increased the extracellular ATP concentration by 11 times and the extracellular HMGB1 concentration by 12.7 times, compared to untreated live cells. Next, irradiated or IRE-treated cells, within 30 min of treatments, were added to bone marrow-derived murine DCs and cultured for 24 h. Flow cytometry showed that 10 Gy-radiated cells were similar with treatment-naive live cells in DC maturation/activation, while IRE-treated cells increased the expression in CD40, CCR7, and CD86 by 72 ± 2%, 52 ± 4%, and 51 ± 4% (mean ± SEM, n = 3), respectively, compared to live cells (Fig. 8d). In vivo radiation was performed using a CT-guided X-ray irradiator to focus the

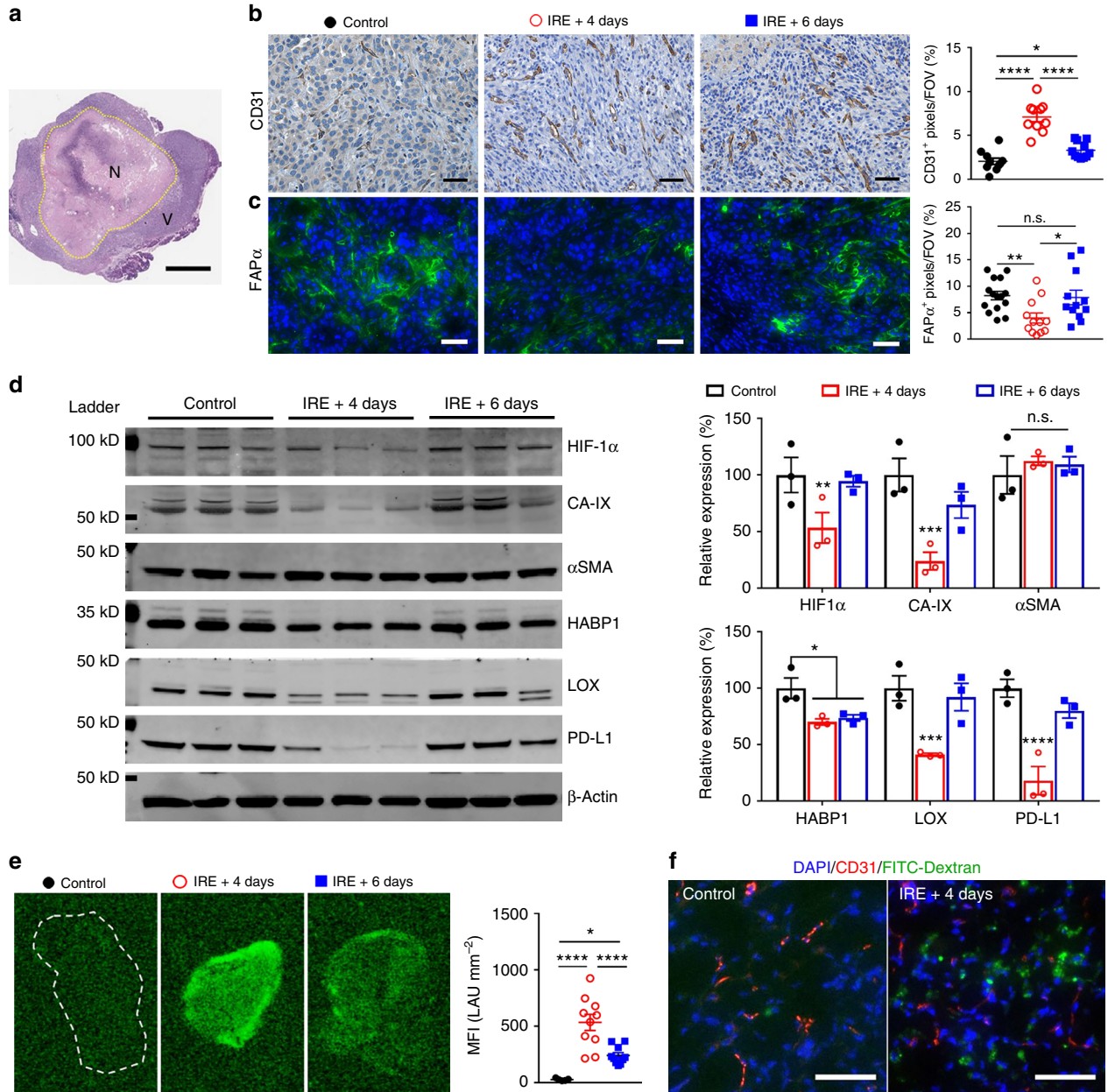

**Fig. 6** Impacts of IRE on PDAC stroma. **a** Representative tumor cross-section after IRE treatment; scale bar = 2 mm. N necrosis, V viable tumor. **b, c** Representative IHC staining of CD31 and FAPαs in viable tumor regions in the groups of control (black solid circle), IRE + 4 days (red circle), and IRE + 6 days (blue solid square), and corresponding quantifications. Ten to fifteen 200× fields-of-view (FOV) were randomly recorded and analyzed. Scale bar = 50 μm. **d** Immunoblotting of tumor lysates (n = 3 per group) for hypoxia-inducible factor 1 alpha (HIF-1α), carbonic anhydrase 9 (CA-IX), αSMA, hyaluronic acid binding protein 1 (HABP1), lysyl oxidase (LOX), PD-L1, and beta-actin (β-actin). **e** Representative fluorescence images of tumor cross-sections and corresponding quantification of mean fluorescence intensity (MFI). Five to ten FOVs were randomly imaged in each group and quantified in terms of MFI. FITC-conjugated dextran (70 kD) was intravenously injected to characterize vascular permeability 24 h before tumors were collected. **f** Representative dual-fluorescence images of CD31 (red) and FITC-dextran (green) in tumors. Scale bar = 50 μm. Data are presented as mean ± SEM. Significance was determined using 1-way ANOVA followed by Tukey post hoc analysis. *$p < 0.05$, ***$p < 0.001$, ****$p < 0.0001$, n.s. not significant. Source data are provided as a Source Data file

radiation dose (10 Gy) at the left abdominal region encompassing pancreas. We limited the radiation dose to 10 Gy because of the potential damage to vital organs surrounding pancreas at higher radiation doses in this orthotopic model. In comparison, a radiation dose of 20 Gy was used to treat subcutaneous PDAC model in another study[33]. Anti-PD1 was injected at the same schedule as that in the IRE + anti-PD1 group. We have shown in Fig. 4a that the long-term surviving mice rejected tumor re-challenge for additional 2 months after

the initial 60-day survival. Therefore, we re-plotted the survival curve for the IRE + anti-PD1 treatment group in Fig. 1a up to 120 days together with that of the 10 Gy + anti-PD1 group (Fig. 8e). The combined 10 Gy + anti-PD1 treatments resulted in a similar median survival compared to IRE + anti-PD1 (30 vs. 31.5 days, $p = 0.16$, log-rank test). However, all mice in the 10 Gy + anti-PD1 experienced tumor progression, and died by 55 days after radiation; while 4 out of the 11 IRE + anti-PD1-treated mice survived for 120 days.

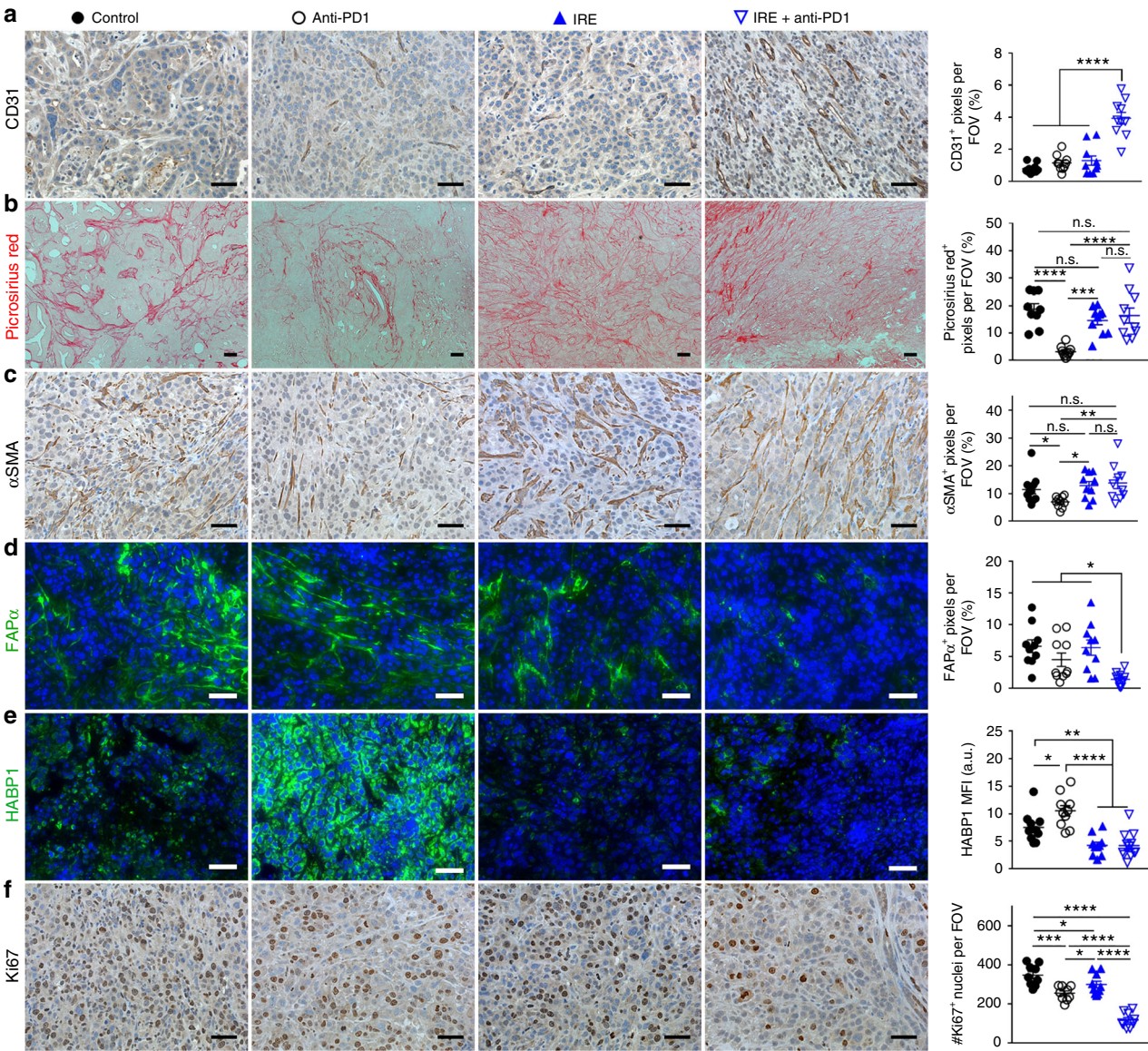

**Fig. 7** IHC staining of viable tumor region at 9 days after initiation of treatments. Representative micrographs of staining for CD31 (**a**), Picrosirius Red (**b**), αSMA (**c**), FAPα (**d**), HABP1 (**e**), and Ki67 (**f**) and corresponding quantifications. Ten to fifteen visual fields were randomly captured each group for control (black solid circle), anti-PD1 (black circle), IRE (blue solid triangle), and IRE + anti-PD1 (blue triangle). Scale bars = 50 μm. Five doses of anti-PD1 had been administered at this time point for the anti-PD1 and combination groups. Data are presented as mean ± SEM. Significance was determined using 1-way ANOVA followed by Tukey post hoc analysis. $*p < 0.05$, $**p < 0.01$, $***p < 0.001$, $****p < 0.0001$. FOV field of view, n.s. not significant. Source data are provided as a Source Data file

To further understand the impact on tumor microenvironment, mice treated with sham control, 10 Gy + anti-PD1, or IRE + anti-PD1 were euthanized at 9 days after radiation or IRE, and analyzed for immune cells and stroma components. The tumor weight was in the order of sham control > 10 Gy + anti-PD1 > IRE + anti-PD1 (Fig. 8f). The intratumoral frequency of CD8+ T cells in the 10 Gy + anti-PD1 group was 2 times that in the sham control, but only half that in the IRE + anti-PD1 group (Fig. 8g). There was no significant difference in the frequency of Tregs (Fig. 8h). The CD8-to-Treg ratio in the IRE + anti-PD1 group was significantly higher than those in the control or 10 Gy + anti-PD1 groups (Fig. 8i). The representative flow cytometry plots and gating strategy are shown in Supplementary Figure 13. Hematoxylin–eosin staining showed that IRE + anti-PD1 induced substantially more necrosis than both sham control and 10 Gy + anti-PD1 (Fig. 8j).

IHC staining of viable tumor regions revealed that 10 Gy + anti-PD1-treated tumors had similar MVD with the sham control, both of which were significantly lower than the IRE + anti-PD1-treated tumors (Fig. 9a). There was no significant difference in the levels of Picrosirius Red staining or αSMA among the three groups (Fig. 9b, c). The levels of FAPα and HABP1 were similar in the groups of sham control and 10 Gy + anti-PD1, both of which were significantly higher than in the IRE + anti-PD1 group (Fig. 9d, e). IRE + anti-PD1 was also more potent than 10 Gy + anti-PD1 in suppressing tumor proliferation (Fig. 9f).

## Discussion
In this study, we showed that combined IRE and anti-PD1 immune checkpoint blockade significantly suppressed tumor growth and prolonged the lives of immunocompetent mice bearing well-established orthotopic PDAC or melanoma

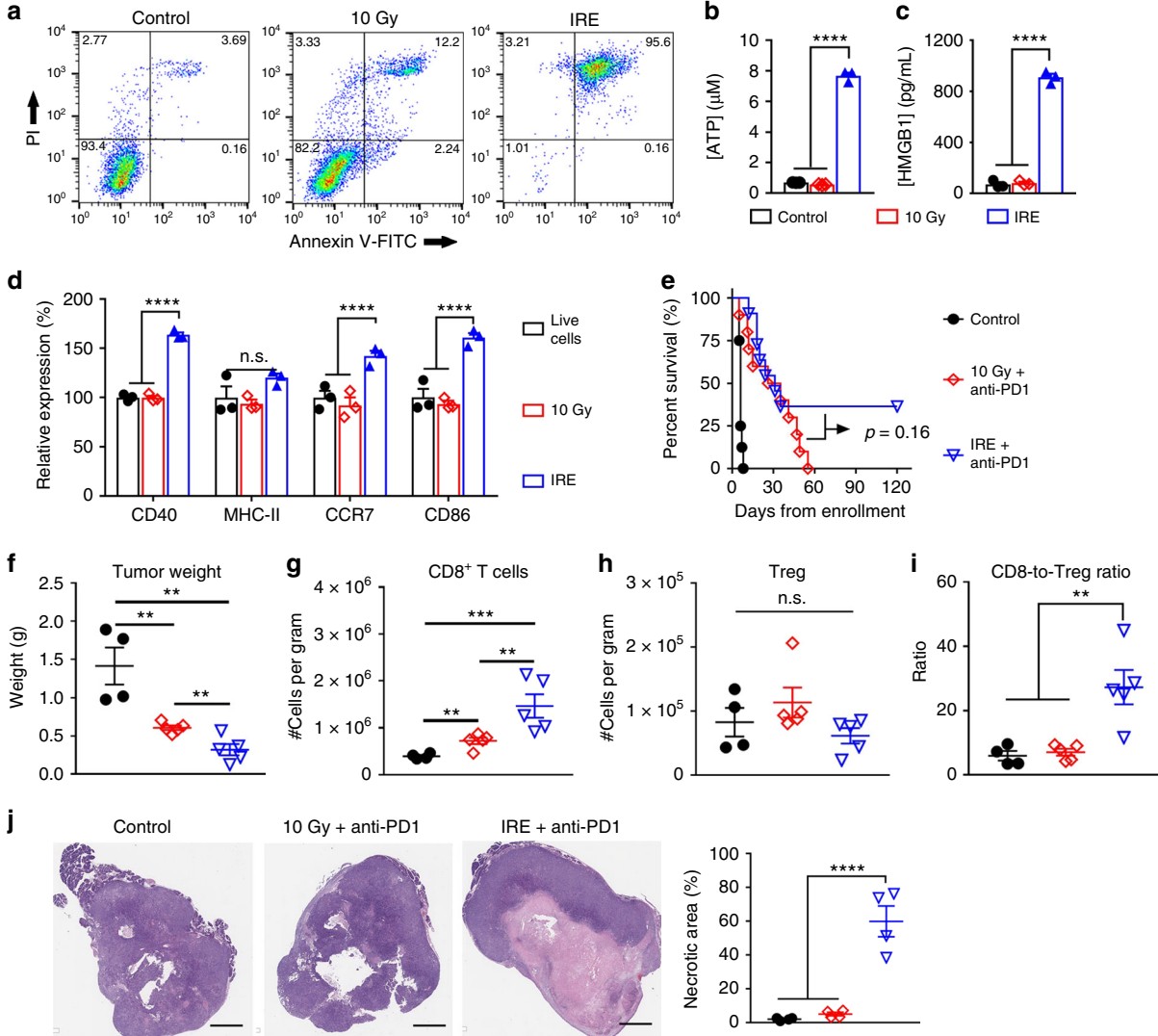

**Fig. 8** Comparison between radiation and IRE in the enhancement of anti-PD1 therapy. **a** Annexin V-FITC/PI staining of KRAS* suspended in phosphate buffered saline (PBS) after 10 Gy of radiation, or electroporated using 960-V, 100-μs electric pulses at 1 Hz frequency for 20 pulses in a 4 mm-gap cuvette. **b**, **c** ATP and HMGB1 concentrations in the cell supernatants of control (black bar), 10 Gy (red bar), and IRE (blue bar) (n = 3). Cells were analyzed within 30 min of treatments. **d** Representative expression of DC activation markers, including CD40, MHC-II, CCR7, and CD86, on bone marrow-derived DCs (CD11c+) after incubation with KRAS* cells for 24 h. Expression was quantified via geometry mean fluorescence intensity (MFI; n = 3). **e** Kaplan–Meier survival curves of mice in sham control (black solid circle, n = 8), 10 Gy + anti-PD1 (red diamond, n = 10), and IRE + anti-PD1 groups (blue triangle, n = 11). **f–i** Tumor weight, frequencies of intratumoral CD8+ T cells and Treg, and CD8-to-Treg ratios from 4 to 6 tumors per group collected at 9 days after radiation or IRE. **j** Representative hematoxylin–eosin staining of tumor sections and quantification of tumor necrosis. Scale bar = 2 mm. Data are presented as mean ± standard error of mean (SEM). Significance was determined using 1-way ANOVA followed by Tukey post hoc analysis or Student's t-test. *p < 0.05, **p < 0.01, ****p < 0.0001, n.s. not significant. Source data are provided as a Source Data file

tumors. Remarkably, in mice with the KRAS* PDAC, 36–43% had a durable response, while in mice with the B16F10 melanoma, 20% had a durable response. Moreover, in the KRAS*-bearing mice that survived for 60 days after the initiation of IRE + anti-PD1 treatment, all rejected tumor cell re-challenge, along with an anti-tumor memory T cell response. Our results demonstrate that tumor-infiltrating CD8+ T cells are the key contributor to the superior anti-tumor efficacy of IRE + anti-PD1. Further mechanistic studies unveiled that the efficacy of IRE + anti-PD1 can be attributed to (1) the activation of DCs that are required to present the tumor-associated antigens to prime T cells, and (2) the IRE-induced alleviation of immunosuppressive components in PDAC stroma, including hypoxia, hyaluronic acid, FAPα, and LOX.

We combined IRE and anti-PD1 to address the lack of immunogenicity[34] and the immunosuppressive tumor micro-environment[2] of PDAC, both of which contribute to the failure of immunotherapy in this disease. IRE enhanced the immunogenicity of KRAS* tumor by releasing DAMPs, which subsequently induced DC activation (Fig. 5b, c). DAMP release during cell killing, including the release of ATP and HMGB1, is an important indicator of immunogenic cell death[35]. ATP functions as a "find me" signal, and can trigger DC activation to prime CD8+ T cell[36]. HMGB1 released into the extracellular matrix activates DCs in a toll-like receptor 4–dependent manner[37]. It is noteworthy that the rate and magnitude of the IRE-induced release of ATP and HMGB1 (about 10 folds in 5 min) were much greater than those by other treatment modalities, such as ionizing radiation and

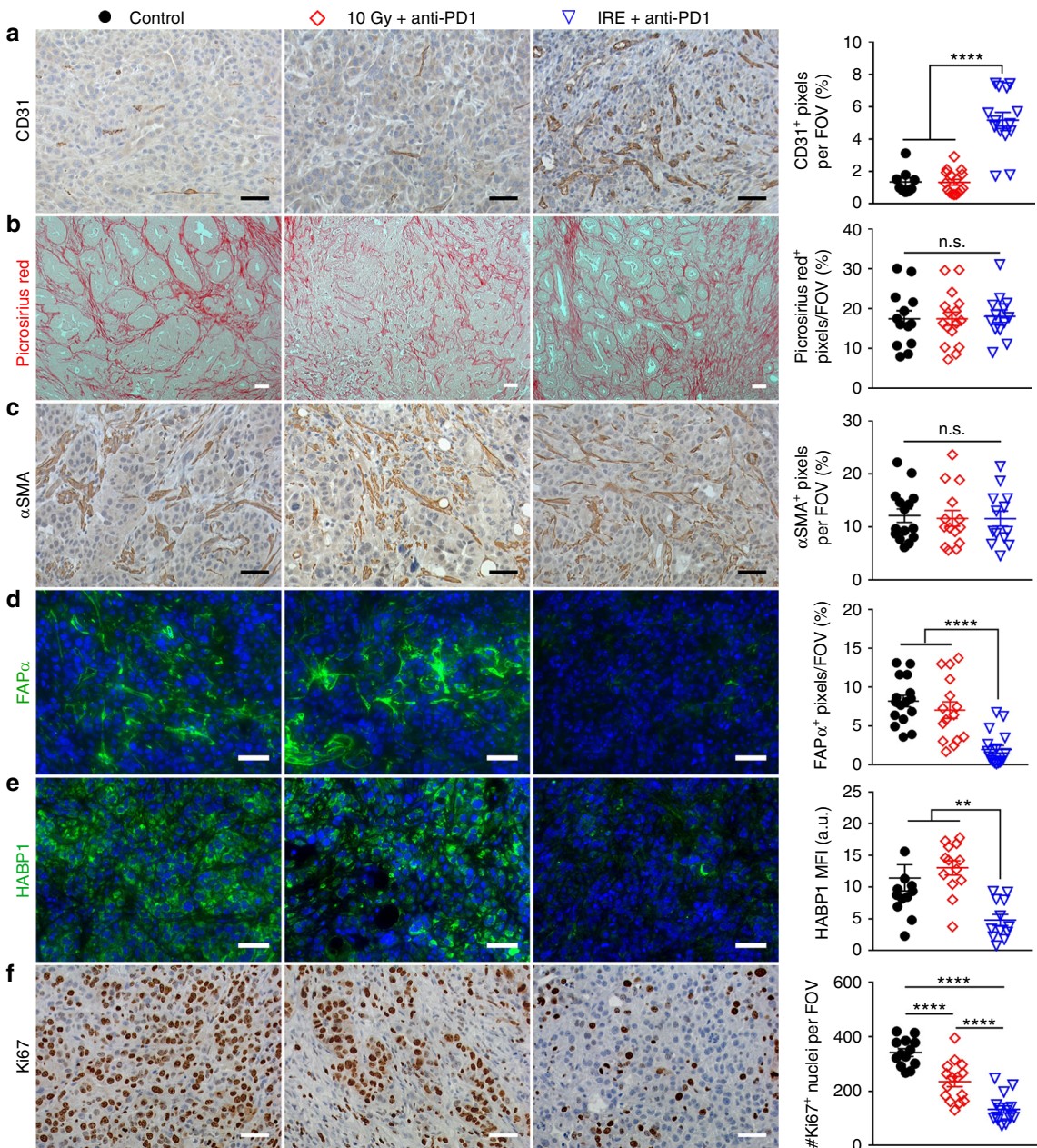

**Fig. 9** IHC staining of viable tumor after 10 Gy + anti-PD1 or IRE + anti-PD1. Representative micrographs of staining for CD31 (**a**), Picrosirius Red (**b**), αSMA (**c**), FAPα (**d**), HABP1 (**e**), and Ki67 (**f**) in control (black solid circle), 10 Gy + anti-PD1 (red diamond), and IRE + anti-PD1 (blue triangle) groups, and corresponding quantifications. Ten to fifteen visual fields were randomly captured. Scale bars = 50 μm. Five doses of anti-PD1 had been administered at this time point for the 10 Gy + anti-PD1 and IRE + anti-PD1 groups. Data are presented as mean ± SEM. Significance was determined using 1-way ANOVA followed by Tukey post hoc analysis. *$p < 0.05$, **$p < 0.01$, ***$p < 0.001$, ****$p < 0.0001$. MFI mean fluorescence intensity, FOV field of view, n.s. not significant. Source data are provided as a Source Data file

chemotherapy (less than 3 folds over several days)[38]. Such an abrupt elevation of DAMP concentrations in the tumor microenvironment may have contributed to the potent anti-tumor immunity in our study.

The immunosuppressive tumor microenvironment was transiently modulated by IRE (Fig. 6), and more persistently modulated by IRE + anti-PD1 (Fig. 7). The increase in both MVD (Figs. 6b and 7a) and permeability of blood vessels (Fig. 6e, f) relieved intratumoral hypoxia, as indicated by the downregulation of HIF-1α and CA-IX (Fig. 6d). Intratumoral hypoxia is a key regulator for immunosuppression[39–42], and reversing hypoxia proved to enhance anti-tumor immunity. For example, breathing 60% oxygen induced a T/NK cell-dependent immune response in lung cancer[43]. On the

other hand, while hyper-oxygenation may cause toxic reactions in airway tissues, IRE selectively relieved hypoxia within tumor stroma (Fig. 6d) without causing significant toxic effects to healthy organs (Supplementary Figure 2). Several components of the fibrotic stroma were also downregulated, including FAPα, hyaluronic acid (indicated by the level of HABP1 expression), and LOX (Fig. 6c, d), all of which can limit the infiltration of cytotoxic T lymphocytes in PDAC. FAPα+ CAFs produce chemokine (C-X-C motif) ligand 12 (CXCL12) that can exclude T cells from cancer cells in PDAC tumors[5]. Depletion of FAPα+ CAFs enhanced anti-PDL1 efficacy in a murine PDAC model[6]. Excessive deposition of hyaluronic acid in PDAC stroma elevates interstitial fluid pressure, which then compresses blood vessels to restrict intratumoral delivery of drugs or

tumor infiltration of immune cells[44]. Combination of hyaluronidase with a *Salmonella*-based immunotherapy increased the accumulation of neutrophils and CD8+ T cells in PDAC tumors and extended animal survival[27]. LOX covalently crosslinks collagen by oxidative deamination of lysine and hydroxylysine residues[28], subsequently producing a fibrotic network that restricts penetration of T cells while supporting the metastatic growth of cancer cells[29,45]. Inhibition of LOX led to stromal depletion and promoted tumor infiltration by drugs and immune cells[46].

Interestingly, the collagen matrix or αSMA+ CAFs were not affected by IRE + anti-PD1 (Fig. 7b, c). Although αSMA+ CAFs contribute to proliferation, metastasis, and clonal expansion of PDAC tumors[47], recent studies revealed that depletion of αSMA+ CAFs led to an aggressive PDAC phenotype[9] and was associated with shorter patient survival in a clinical study[48]. The collagenous matrix of PDAC was also found to restrain, rather than support, PDAC growth[9]. The preservation of extracellular matrix and collagen structures during IRE is consistent with its selective disruption of lipid bilayers[49]. The minimal disruption of collagen matrix and αSMA+ CAFs by IRE + anti-PD1 may have prevented the tumor from an unchecked growth and prolonged the mice's median survival time.

Administration of anti-PD1 after DC activation and stromal modulation promoted selective tumor infiltration by CD8+ and proliferating Ki67+CD8+ T cells (Fig. 3b, c) without recruiting immunosuppressive Tregs (Fig. 3f) and MDSCs (Fig. 3j), therefore produced a favorable immunogenic tumor microenvironment. The higher frequency of memory CD4+ and CD8+ T cells in the spleens of the combination-treated mice (Fig. 4c) indicated that the IRE induced an immunogenic cell death, and the dying tumor cells may have functioned as tumor vaccines to generate a long-term anti-tumor immunity. Indeed, tumor lysate prepared through freeze–thaw cycles is known to immunize against tumor-associated antigens in an unbiased manner[50]. Unlike the laborious ex vivo freeze–thaw procedure, however, IRE can be performed in situ within a short time. The memory immunity was further verified by 100% tumor rejection in our re-challenge study and by the absence of residual tumor in mice that had a durable response after IRE + anti-PD1 therapy (Fig. 4d, Supplementary Figure 10).

Other treatment modalities also have been evaluated along with immunotherapies. For example, gemcitabine was found to enhance immune checkpoint blockade by selectively depleting the immunosuppressive Tregs in PDAC[51]. We previously discovered that the tumor-infiltration by CD8+ T cells could be enhanced by simultaneous stromal modulation and tumor cell killing using nanoparticles co-loaded with cyclopamine and paclitaxel[25]. Ionizing radiation is another commonly used treatment. When combined with anti-PD1 and anti-CTLA4, ionizing radiation eradicated tumor in a murine subcutaneous PDAC model[33]. In our studies, the combination of 10 Gy radiation and anti-PD1 did not modulate the PDAC stroma (Fig. 9), which may account for the lack of long-term survivors. Compared to other local ablation techniques, IRE exhibits several unique advantages. First, the margin of the IRE-ablating zone can be controlled with an accuracy to the thickness of several cells, and therefore can potentially avoid collateral damage to the vital organs surrounding the pancreas[49]. Second, IRE is mostly a non-thermal ablative technique that can preserve the adjacent vessels[52], although thermal damage may occur in the immediate vicinity of electrodes especially when excessive number of electric pulses are delivered[13]. Nevertheless, the preservation of functional blood vessels may have facilitated the infiltration by CD8+ T cells in our study. In contrast, focusing of radiation doses to the deeply seated pancreatic tumor is technically challenging, requiring accurate imaging of the tumor body and strict control of patient body movement[3]. Cryoablation, on the other hand, requires freezing

1 cm beyond the tumor periphery to ensure complete ablation[53], significantly limiting the type of tumors it can be applied to. Future studies should address the complex signaling cascades activated after IRE and the interactions among CAFs, immune cells, and other stromal components. In addition, the use of non-invasive imaging techniques to monitor the molecular changes in PDAC stroma and tumor cells after treatment should be explored.

In conclusion, we report that a novel combination of IRE and anti-PD1 immune checkpoint blockade significantly prolonged survival in mice bearing orthotopic KRAS* PDAC tumors. Given that both IRE and anti-PD1 are already in clinical use, translational studies combining these two treatments in patients with locally advanced PDAC are warranted.

## Methods

**Cell lines and animal models**. All animal studies comply with relevant ethical regulations for animal testing and research, and were approved by the Institutional Animal Care and Use Committee of The University of Texas MD Anderson Cancer Center. Animals were maintained and studies were carried out in accordance with institutional guidelines. B16F10 melanoma cells were obtained from the American Type Culture Collection (ATCC, Manassas, VA). KRAS* cells were kindly provided by Dr. Y Alan Wang at University of Texas MD Anderson Cancer Center. The B16F10 melanoma model was established by subcutaneous inoculation of $5 \times 10^5$ B16F10 murine melanoma cells in 100 μL Hanks balanced salt solution (HBSS) into the lower right back of 6-week-old C57BL/6 mice (Taconic Biosciences, Albany, NY). The KRAS* orthotopic PDAC model was established by intra-pancreatic inoculation of KRAS* murine PDAC cells[54] into 6-week-old C57BL/6 mice. KRAS* cells ($5 \times 10^5$ per mouse) in 10 μL HBSS were injected through a small abdominal incision into the pancreas head. The needle was removed 10 s after completion of the injection, and the incision was closed with absorbable sutures. The cell lines used here were validated by short tandem repeat (STR) DNA fingerprinting by the MD Anderson Cancer Center Characterized Cell Line Core using the AmpFLSTR identifier kit according to manufacturer's instructions (Applied Biosystems, Thermo Scientific, Rockford, IL). The STR profiles were compared to known ATCC fingerprints (ATCC.org), to the Cell Line Integrated Molecular Authentication database, version 0.1.200808 (http://bioinformatics.istge.it/clima/), and to the MD Anderson fingerprint database. The STR profiles matched known DNA fingerprints or were unique. The cells were tested every 2 months for mycoplasma contamination.

**Electroporation**. The experimental procedure was reported following the recommendations by Cemazar et al.[55] for reporting electroporation studies. Briefly, electroporation was performed using the ECM 830 Square Wave Electroporation System (BTX Harvard Apparatus, Holliston, MA). Electroporation parameters, including voltage, pulse duration, pulse repetition frequency, and number of repetition pulses, were set up directly using the control panel of the electroporator. The diagram of pulse sequence is shown in Supplementary Figure 14A. For in vitro experiments, cells were trypsinized, resuspended in phosphate buffered saline (PBS) at $2 \times 10^6$ cells mL$^{-1}$, and added to an electroporation cuvette (Fisher Scientific FB104, Thermo Fisher Scientific, Waltham, MA) embedded with two aluminum plate electrodes 4 mm apart. The cell suspension was in direct contact with the plate electrodes, and subjected to electroporation at room temperature with the following parameters: Voltage: 80–960 V; pulse duration: 100 μs; pulse repetition frequency: 1 Hz; number of repetition pulses: 20. After electroporation, the cell suspension was kept on ice and analyzed or used within 30 min. The cell suspension was subjected to centrifugation at 4 °C, 300g for 5 min. Supernatants were analyzed immediately for ATP measurement or stored at −80 °C for other analyses. Cell pellets were re-suspended in Annexin V binding buffer, stained with Annexin V-FITC/PI (BioLegend, San Diego, CA), and analyzed by flow cytometry (BD FACSCalibur; BD Biosciences, San Jose, CA). For activation of bone marrow-derived DCs, tumor cells were electroporated at $2 \times 10^7$ cells mL$^{-1}$ in PBS, and the whole cell suspension was added to DCs. Three independent repetitions were performed for each in vitro experiment.

Tumor-bearing mice were anesthetized for in vivo IRE experiments. IRE was performed using a 2-needle array electrode with a 5-mm gap made of medical grade stainless steel (BTX item #45-0168, BTX Harvard Apparatus, Holliston, MA). The array was inserted to the center of exposed tumor nodule along the long-axis (Supplementary Figure 14B), and fully penetrated the tumor nodule to maximize the effect of electroporation. The electroporation parameters were: Voltage: 1200 V; pulse duration: 100 μs; pulse repetition frequency: 1 Hz; number of repetition pulses: 99. The incision was closed with absorbable sutures.

**In vivo antitumor efficacy**. Mice with a B16F10 or KRAS* tumor were enrolled in the study once tumor size reached about 7 mm in one dimension. They were randomly assigned to the following groups: (1) untreated control, (2) IRE only, (3) anti-PD1 only, (4) IRE + anti-PD1, (5) anti-CTLA4 + anti-PD1, or (6) IRE + anti-

CTLA4 + anti-PD1. The incision was closed with absorbable sutures. CT-guided X-ray radiation was conducted on an X-RAD SmART small animal imaging-guided irradiation system (Precision X-Ray, North Branford, CT). A total dose of 10 Gy was delivered to the left abdominal region to cover the orthotopic KRAS* tumor. Anti-mouse PD1 antibody (clone J43; Bio X Cell, West Lebanon, NH) and/or anti-mouse CTLA4 antibody (clone 9D9; Bio X Cell) was injected intraperitoneally at 100 μg each per mouse at 30 min after IRE, then every 48 h for 6 total injections. Mice were monitored daily for overall health and euthanized if tumor burden became excessive or the animal became moribund.

**T2-weighted magnetic resonance imaging**. Tumor size was monitored using respiration-gated $T_2$-MRI on a Biospec USR70/30 system (Bruker Biospin MRI, Billerica, MA) equipped with a 7-T magnet. The following parameters were used: TE/TR = 38/2000 ms; BW = 101010.10 Hz; Rare = 8; averages = 3; matrix size = 256 × 192; field of view = 4 cm × 3 cm; slick thickness = 0.75 mm; slice gap = 0.25 mm. Images were processed using Bruker Biospin software. Tumor size was measured at the largest tumor cross-section of axial images.

**Tumor digestion**. Weighed tumors were minced and digested in an 8-mL mixture of 2 mg mL$^{-1}$ collagenase type IV (LS004188; Worthington, Lakewood, NJ), 0.2 mg mL$^{-1}$ hyaluronidase (H3506, Sigma-Aldrich, St. Louis, MO), and 0.2 mg mL$^{-1}$ DNase I (D4527, Sigma-Aldrich) in DMEM/F12 medium at 37 °C for 30 min. The mixture was shaken constantly at 20 RPM. Debris was removed by filtration through a 40-μm mesh, and the red blood cells were removed with a red blood cell lysis buffer (R7757, Sigma-Aldrich). The mixture was pelleted and re-suspended in PBS supplemented with 2% fetal bovine serum for further analyses.

**Analyses of immune cells**. Spleens from treated and untreated tumor-bearing mice were minced and subjected to red blood cell lysis to obtain splenocytes. Cells were stained by using the Live/Dead Fixable Aqua Dead Cell Stain Kit (Invitrogen, Carlsbad, CA) and then incubated with Fc-block (BD Pharmingen, San Jose, CA). Cells were stained with antibodies at 1:100 dilutions or following manufacturer's instructions. CD11b-Pacific Blue (clone M1/79, Cat#101223), Ly6C-PerCP-Cy5.5 (clone AL-21, Cat#560525), Ly6G-PE-Cy7 (clone 1A8, Cat#127617), CD19-FITC (clone 1D3, Cat#152403), CD8α-PE-Cy7 (clone 53–6.7, Cat#100721), F4/80-PE or F4/80-APC-Cy7 (clone BM8, Cat#123109, Cat#123117), CD4-Pacific Blue or CD4-PerCP-Cy5.5 (clone GK1.5, Cat#100427, Cat#100431), NK1.1-APC (clone PK136, Cat#108709), CD11c-APC or CD11c PerCP-Cy5.5 (clone N418, Cat#117309, Cat#117327), CD62L-PE (clone MEL-14 Cat#104407), CD44-FITC (clone IM7, Cat#103021), CD86-PE-Cy7 (clone GL-1, Cat#105013), CD40-APC (clone 3/23, Cat#124611), CCR7-PE (clone 4B12, Cat#120105), MHC-II-FITC (clone M5/114.15.2, Cat#107605), and Ki67-PE-Cy7 (Clone 16AB, Cat#652425) were obtained from BioLegend (San Jose, CA). Foxp3-PE (clone FJK-16s, #12-5773-82) was obtained from eBioscience (Thermo Fisher Scientific, Waltham, MA). Intracellular staining was performed after fixation and permeabilization with an eBioscience Foxp3/Transcription Factor Fixation/Permeabilization kit Samples were analyzed on a BD FACS Canto II cytometer and the data processed with FlowJo (10.0.7) software.

**Measurement of danger-associated molecular patterns**. HMGB1 in the supernatant of treated KRAS* cells was analyzed using an enzyme-linked immunosorbent assay (ELISA) (R&D Systems, MN). ATP concentration was measured using ATPLite bioluminescence kit (Perkin Elmer, Waltham, MA).

**Western blotting**. Cell or tumor lysates were fractioned on NUPAGE 4–12% Bis–Tris gels (Thermo Fisher Scientific) and transferred to a polyvinylidene fluoride membrane (Millipore, Billerica, MA). Membranes were blocked with Odyssey blocking buffer (LI-COR, Lincoln, NE), blotted with the primary antibodies as described in Supplementary Table 1, and visualized using fluorophore-conjugated anti-rabbit or anti-goat IgGs (LI-COR) (1:5000 dilution) on an Odyssey near-infrared fluorescence scanner (LI-COR). The original un-cut western blot images are included in Supplementary Figure 15.

**Immunohistochemical staining**. Tumors and relative organs were harvested, fixed in formalin, and embedded in paraffin before being cut into 4-μm sections. For immunohistochemical analysis, sections were deparaffinized, rehydrated, and subjected to antigen retrieval for 30 min in 10 mM citrate buffer (pH 6) at 95 °C. After antigen retrieval, slides were blocked in Tris-buffered saline solution (pH 7.4) supplemented with 0.1% Tween-20 and 10% goat or donkey serum prior to incubation with primary antibodies overnight at 4 °C. To visualize staining using the 3,3′-diaminobenzidine (DAB) system, slides were washed and incubated with biotinylated anti-rabbit or anti-goat IgG (Vector Laboratories, Burlingame, CA) and streptavidin-conjugated horseradish peroxidase (DAKO, Carpinteria, CA) for 30 min each. A positive reaction was detected by exposure to DAB according to the manufacturer's instructions. Slides were counterstained with hematoxylin and visualized under a bright-field microscope at ×100 or ×200 magnification. For fluorescence visualization, slides were incubated with fluorophore-conjugated anti-rabbit or anti-goat IgGs (Cell Signaling Technology, Beverly, MA) at room temperature for 1 h and counterstained with Hoechst 33342. Slides were visualized using an Axio Observer.Z1 fluorescence microscope (Carl Zeiss Microscopy, Thornwood, NY). Staining was quantified using at least 10 randomly selected 20× fields of view. Control and treated mice within the same experimental set (at least 5 mice per group) were analyzed. All staining was quantified using NIH ImageJ analysis software (http://rsb.info.nih.gov/nih-image/) with the same threshold for each stain. Primary antibodies are listed in Supplementary Table 1. Positive and negative control staining images are shown in Supplementary Figure 16.

**ELISPOT assay**. Lysates of tumor cells were prepared by freezing and thawing. Mouse DCs were generated from bone marrow mononuclear cells by culturing for 7 days in the presence of granulocyte-macrophage colony-stimulating factor[56]. DCs were pulsed with tumor cell lysate for 3 h at 37 °C and then co-incubated with splenocytes overnight at 37 °C. The formed IFN-γ spots were imaged and counted.

**Statistical analysis**. Values are expressed as mean ± SEM. Differences between groups were evaluated by using the Student's $t$-test or one-way analysis of variance followed by post hoc Tukey multiple comparisons. The log-rank test was used in Kaplan–Meier survival analyses. A $p$ value of less than 0.05 was considered statistically significant.

## Data availability
The authors declare that all the data supporting the findings of this study are presented within the article and/or its Supplementary Information Files. The source data underlying Figures 1–9 and Supplementary Figures 1 and 11 are provided with the paper as a Source Data file.

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

## Acknowledgements

The authors thank Kathryn Hale of the Department of Scientific Publications for editing the manuscript. This work was supported in part by the Skip Viragh Foundation, John S. Dunn Foundation, MD Anderson Cancer Center/Emerson Collective Cancer Research Fund, and Gillson Longenbaugh Foundation. The Small Animal Imaging Facility, Research Animal Support Facility, Histology Core Facility, and Immunology Core Facility are supported by a Cancer Center Support Grant from the U.S. National Institutes of Health (P30CA016672). Xiaofei Wen was supported in part by the National Basic Research Program of China (2015CB931800), National Natural Science Foundation of China (81627901), Union for International Cancer Control and China Anti-Cancer Association Chinese Fellowship (CF/16/431486), and Foster Funding of the Fourth Hospital of Harbin Medical University (HYDSYPY201601).

## Author contributions

J.Z., Xiaofei Wen, M.P.M., S.G., B.S., W.P. and C.L. designed the experiment. J.Z., Xiaofei Wen, L.T., T.L., C.X. and Xiaoxia Wen performed experiments and data analysis. J.Z. and C.L. wrote the manuscript.

## Additional information

**Competing interests:** The authors declare no competing interests.

