## [Peer Review File · Nature Communications]

Reviewers' comments:

Reviewer #1 (Remarks to the Author):

In the light of recent regulatory approval of IRE as ablation therapy for pancreatic cancer this report is of great interest and can raise considerable interest. The paper is bringing new results which are important and should be published after revision. In particular I would like to emphasize that reporting of Materials and methods in particular relation to electroporation is not adequate. I strongly suggest to follow recent recommendations that were prepared by a group of experts in the field – see: Čemažar M, Serša G, Frey W, Miklavčič D, Teissié J. Recommendations and requirements for reporting on applications of electric pulse delivery for electroporation of biological samples.

Bioelectrochemistry 122: 69-76, 2018. DOI 10.1016/j.bioelechem.2018.03.005

An important question that needs to be answered is whether IRE in vivo was performed as suboptimal treatment purposely. It is namely obvious from data presented (how IRE was performed as well as results) that IRE alone was not successful in eradicating tumors in animals (see e.g. Fig 1 B and C). It was previously demonstrated that in immunocompetent mice tumor models IRE was perfectly capable of achieving CR.

Further to this it also is not clear how the choice of pulse parameters – in particular pulse amplitude and number of pulses was selected. In in vitro part the authors have used 20 pulses of 500 and 2400 V/cm amplitude whereas in vivo 99 pulses of 2400 V/cm were used. It needs to be emphasized that geometry of electrodes in vitro and in vivo were completely different which means that cells in vitro and in vivo were exposed to different electric fields, and of course to different number of pulses. Since there is no clear discussion and notion of these readers may be misled by the fact the the same number 2400 appears in both cases – but in reality this is pure coincidence. Please report rather voltage amplitude which was applied and not voltage-to-distance ratio which I assume you were trying to report.

I would also like to stress that others have already shown that immune response is important component to IRE and should be acknowledged for that or at least it should be made clear. See e.g.:

Li X, Xu K, Li W, et al. Immunologic response to tumor ablation with irreversible electroporation. PLoS One. 2012;7:e48749.

Neal RE II, Rossmeisl JH Jr, Robertson JL, et al. Improved local and systemic anti-tumor efficacy for irreversible electroporation in immunocompetent versus immunodeficient mice. PLoS One. 2013;8:e64559.

Garcia PA, Kos B, Rossmeisl JHJr, Pavliha D, Miklavčič D. Predictive therapeutic planning for irreversible electroporation treatment of spontaneous malignant glioma.

Med. Phys. 44: 4968-4980, 2017. 2017 Garcia et al DOI 10.1002/mp.12401

Specific comments and suggestions:

Introduction:

Page 3, line 70:

“IRE uses microsecond high-voltage electric pulses to” – since later in Materials section it becomes obvious that pulses are 100 microseconds long it would be perhaps more appropriate to write “IRE uses short high-voltage electric pulses” instead, or simply state the exact duration of pulses

Page3,4, line 70, 71:

Authors write: “IRE uses microsecond high-voltage electric pulses to create irreparable pores in cell membranes, leading to loss of homeostasis and cell death.” Which is not true or at least is it a gross oversimplification. There is vast literature available but none reports on “irreparable pores”. I suggest removing “irreparable” and cite some of the recent review papers in which phenomenon of electroporation is described. Here is a list of few you can choose from:

Rems L, Miklavčič D. Tutorial: Electroporation of cells in complex materials and tissue. J. Appl. Phys. 119: 201101, 2016. DOI 10.1063/1.4949264

Chunlan Jiang, Rafael V. Davalos, and John C. Bischof. A Review of Basic to Clinical Studies of Irreversible Electroporation Therapy. IEEE TRANSACTIONS ON BIOMEDICAL ENGINEERING, VOL. 62, NO. 1, JANUARY 2015: 4-20. DOI 10.1109/TBME.2014.2367543

Yarmush ML, Golberg A, Serša G, Kotnik T, Miklavčič D. Electroporation-based technologies for medicine: principles, applications, and challenges. Annu. Rev. Biomed. Eng. 16: 295-320, 2014. DOI 10.1146/annurev-bioeng-071813-104622

Page 4, lines 75-76: it is not clear how this hypothesis was derived from/developed bysed on »creating irreperable pores in cell membrane«.

Page 7, line 147: at the end of the sentance you quote ref no 16. it is not clear how this reference is relevant to this statement.

Page 7, line 152 and 154: please elaborate/explain why 500 V/cm and 2400 V/cm were selected. Based on preliminary experiments? Literature?

Page 10, Figure 4A: histological section is consistent with IRE being performed suboptimally.

Page 17, Figure 7A: tumor volume growth curves are usually reported in semilogarithmic scale, i.e. volume should be logarithmic. In this way we could also see how treated mice respond

Page 18, line 303: »In this study, we discovered that...» I suggest replacing “discover” with “show” or “demonstrate”

Page 18, line 318: “DAMP-releasing cell death” should probably be “DAMP” ?

Page 19, lines 334-335: authors say: “IRE selectively relieved hypoxia within tumor stroma without causing significant toxic effects to healthy organs (Supplemental Data Fig. S3).” – the referenced figure brings H&E stained histological sections of non-tumor organs, which shows not toxic effect to healthy organs, but I may have missed data evidencing selective relief hypoxia? Please elaborate.

Page 20, lines: 362,363: how is reference 38 supporting the statement: “suggested that the IRE-killed cells may have functioned as tumor vaccines”?

Page 20, line 373: it is true that IRE is(was) considered nonthermal, but it has been demonstrated that at least in immediate vicinity of the electrodes (and in particular with high/excessive number of pulses delivered) the damage is thermal. A word of caution about that would be appropriate in this place. See e.g.: Garcia PA, Davalos RV, Miklavčič D. A numerical investigation of the electric and thermal cell kill distributions in electroporation-based therapies in tissue. PLOS One 9(8): e103083, 2014.

2014 Garcia et al. DOI 10.1371/journal.pone.0103083

Page 21, lines 376-378: does this statement imply that there is no need to treat the safety margin with IRE?

Page 24, lines 465-467: why were of DAMP molecules HMGB1 and ATP selected?

Reviewer #2 (Remarks to the Author):

This study, for the first time, demonstrated electroporation_a currently FDA approved treatment_can reverse the immunosuppressive microenvironment in pancreatic ductal adenocarcinoma (PDAC) by promoting antigen presentation, stroma modulation, enhanced T cells' penetration and so on. In general, the manuscript looks well written and the message conveyed may contribute to the overall knowledge about the treatment of PDAC as a toughest malignancy. Here are several concerns that may help the authors to further clarify their work.

1) It was said when the tumor mass became 8 mm in diameter, therapeutic interventions started. This is workable for subcutaneously implanted tumors, but how could it be possible for tumors implanted at the pancreatic head?

2) The study endpoint was to compare mice survivals among groups, but tumor size as direct evidence should be more considered.

3) In vivo imaging did not seem to have been regularly applied for monitoring tumor growth and evaluating therapeutic effects.

4) Survival of the control animals appeared exceptionally short, i.e. within 10 days after treatment started when the tumor sized about 8 mm in diameter.

5) Ref 44 cited for modeling PDAC does not seem to be relevant, i.e. the nude mice vs. C57BL/6 mice and xenograph vs. allograph used here. How pancreatic cancer model was created should be described.

Other technical points be to be considered are:

1) Concerning the IHC staining, positive control and negative control are needed to exclude false negative and false positive.

2) In figure 4C, in the result of WB of CA-IX, lanes in the control, as well as in the IRE + 6Days group, seems to be in the cutting margin of members. This may lead to miscalculation of staining area. And the lanes in the WB of CA-IX are not located in one line. Additionally, the quantification and statistical analyses of WB results in figure 4C may demonstrate the difference between groups in a more precise manner.

3) On page 7 line 157, there appear to be two pitfalls in the comparison of DC activation markers between IRE-treated group and PBS group. First, the better control group against IRE-treated cells may be cells which did not be treated by IRE. Second, in the legend of figure 3C, does the "CCD7" mean CCR7?

4) In the figure 6E, the cell number of CD4+ T cells in control group is similar with that in the IRE+anti-PD-1 group. However, in the figure 7C, the percentage of CD4E (although statistically

insignificant) and CD4M in treated group is higher than that in control group. Is there any difference in the measuring methods? Otherwise, what is the cause for this discrepancy? Please clarify this.

5) In the method section on page 23 line 430, PD-1 or CTLA-4 antibody is administered intraperitoneally, which is different from the way in clinic and the instruction of clinically used PD-1 antibody product. And this may create gap in the translation of the promising results in this study. Why not choose intravenous injection, for example, tail vein injection?

6) On page 26 line 273, when demonstrating the long term T cells, the author used treatment-naive healthy mice as control. The better control group here should be that age-matched healthy mice went through sham surgery in the same time before tumor cell re-challenge as treated group.

7) The figure 1A seems not have been cited throughout the manuscript.

8) On page 19 line 340, excessive deposition of hyaluronic acid in PDAC stroma elevates interstitial fluid pressure, which then compresses blood vessels to restrict intratumoral delivery of drugs or immune cells. A citation is needed to support this idea, as it cannot be interpreted from results.

Reviewer #3 (Remarks to the Author):

The manuscript by Zhao et al accesses the impact of irreversible electroporation on immune checkpoint blockade in animal models. The trust of the paper is that irreversible electroporation induces immunogenic cell death and changes the TME to facilitate response to PD1 checkpoint therapy. While a very interesting observation with significant efficacy observed in the animal models run, I find several aspects of the paper that diminish my enthusiasm in the current form.

Major points.

1. The largest aspect of this paper that is troubling, is it is mostly observational in nature and not mechanistic. I find no single set of data that firmly explains why IRE improves response to PD1. There is some data for immunogenic cell death(in-vitro) and some data for TME modulation, but neither are formally linked(with data) to the improvement in response. Thus, these things happen, but what is there impact or IRE that unlocks immunotherapy? This significantly impacts my enthusiasm.

2. Is IRE superior or equivalent to other mediators of cell death, like chemotherapy or radiation in it's ability to improve PD1 response. How does it compare, and what aspects of cell death, T cell biology and TME modulators are similar and different?

3. There are aspects of the data presented that should be improved.

a. The use of western blots, as in Figure 4c to show changes in the TME like FAP+ CAFs is not the best method, as it does not take into account cell diversity. Indeed the authors use IHC in Figure 5 and should in the time course analysis in Figure 4.

b. MFI analysis is not likely appropriate for analysis of FAPa CAF staining shown in Figure 5D. Most investigator show area or cell number (this may just be typo).

c. Did the investigators count both viable and non-viable areas in quantitation in Figure 5. And should they not also show data excluding the necrotic area.

d. The flowcytometry data in figure 3a is not properly compensated. The diagonal staining profile is indicative of this and this aspect would effect the data in all panels. Also was this done in replicates with graphical or merely one time one well/condition. Outcome is believable, but data should be improved.

e. The number of repeat experiments done for each figure panel is not noted. Example “representative of 3 independent invitro experiments”. Thus it’s unclear if the data represent a single outcome or robust repeatable findings. Likely the later but this should be spelled out.

4. Why does PD1 work and CTLA4 not work? Is PDL1 more relevant to IRE and why? Or is CTLA4 not relevant to this mouse model (possible)

Point-to-Point Responses to Reviewers' Comments

Comments from Editor.

(1) We hope you will find the referees' comments useful as you decide how to proceed. Should further experimental data or analysis allow you to address these criticisms, we would be happy to look at a substantially revised manuscript. However, please bear in mind that we will be reluctant to approach the referees again in the absence of major revisions. In particular, we suggest that you include tumor sizes/growth curves and longer survival data in your in vivo experiments and further elucidate the mechanism through which IRE improves response to immunotherapy. If the revision process takes significantly longer than three months, we will be happy to reconsider your paper at a later date, as long as nothing similar has been accepted for publication at Nature Communications or published elsewhere in the meantime.

Answer:

We are grateful for this opportunity to further improve our manuscript. We have performed additional experiments per reviewers' suggestions, and revised the manuscript accordingly. Specifically: (1) MRI images were acquired to monitor tumor sizes and growth. (2) Mice that survived for up to 9 months were analyzed to confirm absence of microscopic tumor nodules. (3) Antibody depletion studies were performed to provide a direct evidence on the critical role of CD8⁺ T cells in the anti-tumor efficacy. (4) Side-by-side comparison between IRE + anti-PD1 and 10 Gy radiation + anti-PD1 was performed to demonstrate the unique outcomes with the combined IRE + anti-PD1 therapy.

(2) When resubmitting your paper, please highlight all changes in the manuscript text file. We also ask that you ensure that your manuscript complies with our editorial policies. Specifically, please ensure that the following requirements are met, and any relevant checklists are completed or updated and uploaded as a Related Manuscript file type with the revised article:

To improve the quality of methods and statistics reporting in our papers, we are now asking all authors to complete an editorial policy checklist that verifies compliance with all required editorial policies. Please ensure that the checklist is completed and uploaded with your revised article. All points on the policy checklist must be addressed; if needed, please revise your manuscript in response to these points. Please note that this form is a dynamic 'smart pdf' and must therefore be downloaded and completed in Adobe Reader.

Editorial policy checklist: <https://www.nature.com/authors/policies/Policy.pdf>

Answer:

All changes are highlighted. Manuscript was reformatted to comply with editorial policies. All authors have signed the form of editorial policy. The signed documents and reporting summary are uploaded.

Comments from Reviewer #1

In the light of recent regulatory approval of IRE as ablation therapy for pancreatic cancer this report is of great interest and can raise considerable interest. The paper is bringing new results which are important and should be published after revision.

Answer: We thank the reviewer for the encouragement. We are currently working with the physicians in the Departments of Interventional Radiology and Surgical Oncology at MD Anderson Cancer Center to develop protocols for potential clinical trial studies. We hope that a timely reporting of our results in highly visible journals, e.g. Nature Communication, will help accelerate the clinical translation, and make a significant impact on the therapeutic outcome of PDAC patients. Detailed responses are listed as follows.

(1) In particular I would like to emphasize that reporting of Materials and methods in particular relation to electroporation is not adequate. I strongly suggest to follow recent recommendations that were prepared by a group of experts in the field – see: Čemažar M, Serša G, Frey W, Miklavčič D, Teissié J. Recommendations and requirements for reporting on applications of electric pulse delivery for electroporation of biological samples. *Bioelectrochemistry* 122: 69-76, 2018. DOI 10.1016/j.bioelechem.2018.03.005

Answer:

Detailed description of the electroporation procedure was added to the Materials and Methods part following the guidelines by **Cemazar et al.**, which was also added to the reference list as Reference 55.

Materials and Methods-Electroporation

The experimental procedure was reported following the recommendations by Cemazar et al.⁵⁵ for reporting electroporation studies. Briefly, electroporation was performed using the ECM 830 Square Wave Electroporation System (BTX Harvard Apparatus, Holliston, MA). Electroporation parameters, including voltage, pulse duration, pulse repetition frequency, and number of repetition pulses, were set up directly using the control panel of the electroporator. The diagram of pulse sequence is shown in **Supplemental Data Fig. S14A**.

For in vitro experiments, cells were trypsinized, resuspended in phosphate buffered saline (PBS) at 2×10^6 cells/mL, and added to an electroporation cuvette (Fisher Scientific FB104, Thermo Fisher Scientific, Waltham, MA) embedded with two aluminum plate electrodes 4 mm apart. The cell suspension was in direct contact with the plate electrodes, and subjected to electroporation at room temperature with the following parameters: Voltage: 80 to 960 V; pulse duration: 100 μ second; pulse repetition frequency: 1 Hz; number of repetition pulses: 20. After electroporation, the cell suspension was kept on ice and analyzed or used within 30 minutes. The whole cell suspension was used for activation of bone marrow-derived dendritic cells. Otherwise, the cell suspension was subjected to centrifugation at 4 °C, 300 g for 5 minutes. Supernatants were analyzed immediately for ATP measurement or stored at -80 °C for other analyses. Cell pellets were re-suspended in Annexin V binding buffer, stained with Annexin V-FITC/PI (BioLegend, San Diego, CA), and analyzed by flow cytometry (BD FACSCalibur; BD Biosciences, San Jose, CA). Three independent repetitions were performed for each in vitro experiment.

Tumor-bearing mice were anaesthetized for *in vivo* IRE experiments. IRE was performed using a 2-needle array electrode with a 5-mm gap made of medical grade stainless steel (BTX item #45-0168, BTX Harvard Apparatus, Holliston, MA). The array was inserted to the center of exposed tumor nodule along the long-axis (shown in **Supplemental Data Fig. S14B**), and fully penetrated the tumor nodule to maximize the effect of electroporation. The electroporation parameters were: Voltage: 1200 V; pulse duration: 100 μ second; pulse repetition frequency: 1 Hz; number of repetition pulses: 99. Three to eleven mice were enrolled for each

(2) An important question that needs to be answered is whether IRE in vivo was performed as suboptimal treatment purposely. It is namely obvious from data presented (how IRE was performed as well as results) that IRE alone was not successful in eradicating tumors in animals (see e.g. Fig 1 B and C). It was previously demonstrated that in immunocompetent mice tumor models IRE was perfectly capable of achieving CR.

Answer:

We did not contrive to achieve a complete tumor eradication with IRE alone, because the focus of this study was to demonstrate the efficacy and feasibility of combining IRE and anti-PD1 immunotherapy.

The reviewer is correct in that the complete response by IRE is possible in animal models. However, clinical trials with IRE or IRE + chemo/radiation in human patients with pancreatic cancer has not significantly increased the long-term survival (*Ann Surg Oncol* 2013, 20 Suppl 3, S443-449). These clinical findings indicated that it would be difficult to achieve a complete response by IRE alone or in combination with chemo/radiotherapy in human patients. In addition, recent studies (*Radiology* 2015, 274, 115-123; *Sci Rep* 2015, 5, 8485) have shown that the strength of electric field around IRE probes was inhomogeneous, and there were regions where tumor cells only experienced reversible electroporation. Such suboptimal IRE treatment may lead to more aggressive tumors than un-treated ones (*Mol Ther Methods Clin Dev* 2015, 2, 15001). Therefore, there is an urgent medical need to treat these residual tumors, a major focus of current study.

(3) Further to this it also is not clear how the choice of pulse parameters – in particular pulse amplitude and number of pulses was selected. In in vitro part the authors have used 20 pulses of 500 and 2400 V/cm amplitude whereas in vivo 99 pulses of 2400 V/cm were used. It needs to be emphasized that geometry of electrodes in vitro and in vivo were completely different which means that cells in vitro and in vivo were exposed to different electric fields, and of course to different number of pulses. Since there is no clear discussion and notion of these readers may be misleading by the fact the same number 2400 appears in both cases – but in reality this is pure coincidence. Please report rather voltage amplitude which was applied and not voltage-to- distance ratio which I assume you were trying to report.

Answer:

We apologize for the confusions. The updated experimental procedures are now added per reviewer's suggestions. IRE parameters are reported now as voltage instead of V/cm. Actual voltage amplitude and geometry of electrodes were reported for both *in vitro* and *in vivo* settings throughout the manuscript. The reviewer is correct that the 2400 V/cm is a pure coincidence. We chose 20 pulses for in vitro studies to minimize the thermal effect. The number of 99 pulses was the maximal number allowed by the electroporator for a single run. We chose it to simply the experiment procedure.

(4) I would also like to stress that others have already shown that immune response is important component to IRE and should be acknowledged for that or at least should be made clear. See e.g.: Li X, Xu K, Li W, et al. Immunologic response to tumor ablation with irreversible electroporation. *PLoS One*. 2012;7:e48749. Neal RE II, Rossmeisl JH Jr, Robertson JL, et al. Improved local and systemic anti-tumor efficacy for irreversible electroporation in immunocompetent versus immunodeficient mice. *PLoS One*. 2013;8: e64559. Garcia PA, Kos B, Rossmeisl JHJr, Pavliha D, Miklavčič D. Predictive therapeutic planning for irreversible

electroporation treatment of spontaneous malignant glioma. *Med. Phys.* 44: 4968-4980, 2017. 2017 Garcia et al DOI 10.1002/mp.12401

Answer:

We thank reviewer #1 for the input. We have cited relevant references in the revised manuscript, listed below:

On the other hand, recent studies on other tumor models, including a rat sarcoma,¹⁹ a murine renal carcinoma,²⁰ and a canine glioma model,²¹ have shown an improved antitumor efficacy of IRE in immunocompetent animals, suggesting a possible role of the host immune system in mediating antitumor effects of IRE.

Specific comments and suggestions:

Introduction:

(1) Page 3, line 70:

"IRE uses microsecond high-voltage electric pulses to" – since later in Materials section it becomes obvious that pulses are 100 microseconds long it would be perhaps more appropriate to write "IRE uses short high-voltage electric pulses" instead, or simply state the exact duration of pulses.

Answer:

As suggested, the sentence was changed to "IRE uses short high-voltage electric pulses..."

(2) Page3, 4, line 70, 71:

Authors write: "IRE uses microsecond high-voltage electric pulses to create irreparable pores in cell membranes, leading to loss of homeostasis and cell death." Which is not true or at least is it a gross oversimplification. There is vast literature available but none reports on "irreparable pores". I suggest removing "irreparable" and cite some of the recent review papers in which phenomenon of electroporation is described. Here is a list of few you can choose from:

Rems L, Miklavčič D. Tutorial: Electroporation of cells in complex materials and tissue. *J. Appl. Phys.* 119: 201101, 2016. DOI 10.1063/1.4949264

Chunlan Jiang, Rafael V. Davalos, and John C. Bischof. A Review of Basic to Clinical Studies of Irreversible Electroporation Therapy. *IEEE TRANSACTIONS ON BIOMEDICAL ENGINEERING, VOL. 62, NO. 1, JANUARY 2015: 4-20.* DOI 10.1109/TBME.2014.2367543

Yarmush ML, Golberg A, Serša G, Kotnik T, Miklavčič D. Electroporation-based technologies for medicine: principles, applications, and challenges. *Annu. Rev. Biomed. Eng.* 16: 295-320, 2014. DOI 10.1146/annurev-bioeng-071813-104622

Answer:

We agree that our previous statement about "irreparable pores" was over-simplified. Based on the summary from the review papers (cited in the revised manuscript), the paragraph was changed to the following:

"IRE uses short high-voltage electric pulses to induce cell death through permanent membrane lysis or loss of homeostasis.^{15-17"}

(3) Page 4, lines 75-76: it is not clear how this hypothesis was derived from/developed based on »creating irreparable pores in cell membrane«.

Answer:

The paragraph was re-written to improve the logic flow as follows:

In PDAC, IRE has been shown to increase tumor delivery of gemcitabine,¹⁸ indicating a potential modulation of the stroma. However, the exact extent of IRE-induced stromal change remains unknown. On the other hand, recent studies on other tumor models, including rat sarcoma,¹⁹ murine renal carcinoma,²⁰ and canine glioma,²¹ have shown an improved antitumor efficacy of IRE in immunocompetent animals, suggesting a possible role of the host immune system in mediating antitumor effects of IRE. However, these studies were not performed in the context of immunotherapy. Neither did the studies involve stromal modulation. Up to date, it is unknown whether IRE can potentiate the antitumor efficacy of immunotherapy in poorly immunogenic PDAC.

Based on these analyses, we hypothesized that IRE enhances the efficacy of anti-PD1 therapy by activating the immune system and alleviating stroma-induced immunosuppression in PDAC.

(4) Page 7, line 147: at the end of the sentence you quote ref no 16. it is not clear how this reference is relevant to this statement.

Answer:

Reference 16 in the original manuscript has been removed.

(5) Page 7, line 152 and 154: please elaborate/explain why 500 V/cm and 2400 V/cm were selected. Based on preliminary experiments? Literature?

Answer:

The voltages for reversible and irreversible electroporation *in vitro* were determined by a pilot study (**Supplemental Data Fig. S11**). For *in vitro* setting in a 4-mm gap cuvette, voltage below 200 V did not significantly affected KRAS* cell viability, while the voltage above 480 V caused significant damage to cells.

(6) Page 10, Figure 4A: histological section is consistent with IRE being performed sub-optimally.

Answer:

Yes, we did not contrive to achieve complete response with IRE alone. The figure is re-arranged as **New Fig. 6A**.

(7) Page 17, Figure 7A: tumor volume growth curves are usually reported in semi-logarithmic scale, i.e. volume should be logarithmic. In this way we could also see how treated mice respond

Answer:

The tumor volume growth curve (**New Fig. 4A**) is changed to semi-logarithmic. The treated mice did not exhibit any tumor growth. The original inoculation site had a small bump (about 4 mm in diameter) due to the volume of injected cell suspension. The bump was quickly absorbed in a few days, leaving a small skin bump of about 1 mm in diameter, and never grew any larger. Therefore, the volume for treated mice is assigned as 1 mm³ in order to be shown on the logarithmic scale.

New Fig. 4A. Growth curve of rechallenged tumors in the long-term surviving mice treated with IRE + anti-PD1 (n = 10). Age-matched healthy mice that went through sham surgery at the same time of the treated mice were used as controls (n = 9). Time point of euthanasia is indicated by an arrow. Data presented as interquartile range (IQR) with a median center line and min to max error bars. There was no growth of rechallenged tumor in the long-term surviving mice. Their volume was assigned to be 1 mm³ in order to be visible on the semi-logarithmic scale.

(8) Page 18, line 303: »In this study, we discovered that...» I suggest replacing “discover” with “show” or “demonstrate”

Answer:

"Discovered" is changed to "showed".

(9) Page 18, line 318: “DAMP-releasing cell death” should probably be “DAMP”?

Answer:

The sentence is changed to "IRE, but not ionizing radiation, enhanced the immunogenicity of KRAS* tumor by releasing DAMPs, which subsequently induced DC activation (**new Fig. 5**)."

(10) Page 19, lines 334-335: authors say: “IRE selectively relieved hypoxia within tumor stroma without causing significant toxic effects to healthy organs (Supplemental Data Fig. S3).” – the referenced figure brings H&E stained histological sections of non-tumor organs, which shows not toxic effect to healthy organs, but I may have missed data evidencing selective relief hypoxia? Please elaborate.

Answer:

We apologize for the confusion. The selective relief of tumor hypoxia was presented in **new Fig. 6D**, where the western blots showed a decrease in hypoxia markers (HIF-1 α and CA-IX) at 4 days after IRE.

The paragraph is changed to "IRE selectively relieved hypoxia within tumor stroma (**Fig. 6D**) without causing significant toxic effects to healthy organs (**Supplemental Data Fig. S2**)."

(11) Page 20, lines: 362,363: how is reference 38 supporting the statement: “suggested that the IRE-killed cells may have functioned as tumor vaccines”?

Answer:

Reference 38 in the original manuscript was removed. Our data of memory T cells suggested that IRE-killed cells may have provided antigens and functioned as tumor vaccines.

(12) Page 20, line 373: it is true that IRE is(was) considered nonthermal, but is has been demonstrated that at least in immediate vicinity of the electrodes (and in particular with high/excessive number of pulses

delivered) the damage is thermal. A word of caution about that would be appropriate in this place. See e.g.: Garcia PA, Davalos RV, Miklavčič D. A numerical investigation of the electric and thermal cell kill distributions in electroporation-based therapies in tissue. PLOS One 9(8): e103083, 2014. 2014 Garcia et al. DOI 10.1371/journal.pone.0103083

Answer:

We thank the reviewer for the clarification. The reference by *Garcia PA et al.* is cited in the revised manuscript, and the corresponding discussion is revised as follow:

Second, IRE is mostly a non-thermal ablative technique that can preserve the adjacent vessels,⁵² although thermal damage may occur in the immediate vicinity of electrodes especially when excessive number of electric pulses are delivered.¹³ Nevertheless, the preservation of functional blood vessels may have facilitated the infiltration by CD8⁺ T cells in our study.

(13) Page 21, lines 376-378: does this statement imply that there is no need to treat the safety margin with IRE?

Answer:

No. The statement was meant to show that cryoablation may not be suitable for treating tumors that are in direct contact with vital organs. We acknowledge that IRE is useful for margin accentuation during resection of border-line resectable PDAC (*Med Biol Eng Comput.* 2017 Jul;55(7):1123-1127).

(14) Page 24, lines 465-467: why were of DAMP molecules HMGB1 and ATP selected?

Answer:

Representative DAMPs and their roles in immune activation are reviewed by *Krysko, D. V. et al.* in "Immunogenic cell death and DAMPs in cancer therapy, *Nat. Rev. Cancer* 12, 860-875, (2012)." In addition to ATP and HMGB1, other DAMPs include calreticulin, heat-shock proteins, and histones, etc. We measured ATP and HMGB1 as a proof-of-concept experiment to demonstrate that IRE induced release of intracellular DAMPs. Detailed study on DAMPs related to IRE would require a separate study beyond the scope of this manuscript.

Comments from Reviewer #2

This study, for the first time, demonstrated electroporation, a currently FDA approved treatment can reverse the immunosuppressive microenvironment in pancreatic ductal adenocarcinoma (PDAC) by promoting antigen presentation, stroma modulation, enhanced T cells' penetration and so on. In general, the manuscript looks well written and the message conveyed may contribute to the overall knowledge about the treatment of PDAC as a toughest malignancy. Here are several concerns that may help the authors to further clarify their work.

Answer:

We thank the reviewer for the positive evaluation. Revisions are made according to the reviewer's suggestions. We are currently working with the physicians in the Departments of Interventional Radiology and Surgical Oncology at MD Anderson Cancer Center to develop protocols for potential clinical evaluation. We hope that reporting our results in a highly visible journal, e.g. Nature Communication, would accelerate the clinical translation of this innovative combinatory therapy strategy to benefit PDAC patients.

1) It was said when the tumor mass became 8 mm in diameter, therapeutic interventions started. This is workable for subcutaneously implanted tumors, but how could it be possible for tumors implanted at the pancreatic head?

Answer:

We apologize for the confusion. The reviewer is correct that it is hard to directly measure the size of orthotopic pancreatic tumor. The size requirement of tumor at the time of IRE was determined by the two-needle array, which had a gap of 5 mm. We chose this array system based on its similarity with the clinically approved NanoKnife system for IRE. As shown in **Supplemental Data Fig. S14B**, the two needles were inserted at the center of the tumor nodule along the long-axis, and fully penetrated the whole nodule. In practice, we palpate the solid tumor nodule after anesthetizing the mice (for about 2 minutes). By palpation, we could determine if the tumor length could accommodate the two-needle array. Then we performed an open surgery (**Fig. 1A**) to expose the tumor nodule for IRE treatment (or sham surgery). At this step, we were able to accurately measure the tumor size with a caliper to ensure that tumor sizes were consistent among all the treatment groups. We agree that *in vivo* imaging is a more accurate method to measure tumor size.

2) The study endpoint was to compare mice survivals among groups, but tumor size as direct evidence should be more considered.

Answer:

Per reviewer's suggestions, we performed an additional small-scale study and used T₂-weighted MRI to monitor tumor growth in the 4 treatment groups. Three mice were enrolled in each of the sham control, anti-PD1, and IRE groups, and 5 mice were enrolled in the IRE + anti-PD1 group. The representative results are summarized in **Fig. 1D** and the whole imaging set are provided in **Supplemental Data Figs. S3-6**. All mice in the control, anti-PD1, and IRE groups died of excessive tumor burden, while 2 out of the 5 IRE + anti-PD1-treated mice showed no tumors on MRI images at 42 days after initiation of treatments. Therefore, these imaging results were consistent with the survival data.

Fig. 1D. Axial T₂-MRI images of one mouse from each group. Mice were enrolled for treatment once the tumor size reached about 7 mm in one dimension (assigned as day 0). MRI slice with the largest tumor cross-section is presented to demonstrate tumor size for each time point. MRI images were scanned weekly until the mice were euthanized due to excessive tumor burden.

Fig. S3. T₂-MRI images of sham control KRAS* tumor and corresponding photographs. Three mice were imaged. Scale bar = 5 mm.

Fig. S4. T₂-MRI images of anti-PD1-treated KRAS* tumor and corresponding photographs. Anti-PD1 was intraperitoneally injected at 100 µg per mouse starting from day 0, then every 48 hours for 6 total doses. Three mice were imaged. Scale bar = 5 mm.

Fig. S5. T₂-MRI images of anti-PD1-treated KRAS* tumor and corresponding photographs. IRE was conducted using a two-electrode array with 5 mm between the electrodes. The array was inserted into the tumor center. The parameters of electric pulses were: 1200 V, 100 μ second duration per pulse, 1 pulse per second, 99 pulses in total. Three mice were imaged. Scale bar = 5 mm.

Fig. S6. T₂-MRI images of IRE + anti-PD1-treated KRAS* tumor and corresponding photographs. IRE was conducted using a two-electrode array with 5 mm between the electrodes. The array was inserted into the tumor center. The parameters of electric pulses were: 1200 V, 100 μ second duration per pulse, 1 pulse per second, 99 pulses in total. Anti-PD1 was intraperitoneally injected at 100 μ g per mouse starting from day 0, then every 48 hours for 6 total doses. Five mice were imaged. Scale bar = 5 mm.

3) In vivo imaging did not seem to have been regularly applied for monitoring tumor growth and evaluating therapeutic effects.

Answer:

We performed a small-scale study to image KRAS* tumor growth with T₂-MRI imaging (results shown above). We would like to point out that animal survival is an established and widely-used method to evaluate therapeutic effects (*Cancer Cell. 2014 Jun 16; 25(6): 719–734*). The following technical difficulties prevented us from using T₂-MRI imaging to routinely monitor all the mice in survival study.

(1) MRI imaging is slow, and requires about 15 minutes for each mouse. All whole set of 40 mice in the survival study would need 10 hours for each scan. Due to the limited availability of MRI scanner and the high cost associated with imaging study, we were only able to perform a small-scale study, and only scanned the mice once per week.

(2) Mice were respiration-gated during MRI scan. Their breathing rate was controlled to be below 25 breaths per minute by adjusting the anesthesia level (isoflurane concentration). This was a severe stress for tumor-bearing mice, especially for the mice with late-stage tumor. In pilot studies, we had lost several mice due to

anesthesia. Therefore, we preferred not to anesthetize the mice in survival study for any extended time. In contrast, they only needed to be anesthetized for about 2 minutes for palpating the tumor size.

4) Survival of the control animals appeared exceptionally short, i.e. within 10 days after treatment started when the tumor sized about 8 mm in diameter.

Answer:

Yes. KRAS* tumor was a highly aggressive model, in part due to its orthotopic location. The MRI images (shown above) have demonstrated that sham control tumor grew to about 20 mm in length within about 9 days. The criteria of excessive tumor burden for orthotopic tumors is 20 mm in one dimension according to institutional guidelines.

5) Ref 44 cited for modeling PDAC does not seem to be relevant, i.e. the nude mice vs. C57BL/6 mice and xenograft vs. allograft used here. How pancreatic cancer model was created should be described.

Answer:

KRAS* model was established by intra-pancreatic inoculation of KRAS* cells with a doxycycline-inducible mutation of KRASG12D. The generation of KRAS* cells were described in the original **Reference 44 (New Reference 54, *Cell. 2012 Apr 27;149 (3):656-70*)**. The procedure of inoculation was described in the animal model section:

KRAS* Cells (5×10^5 per mouse) in 10 μ L HBSS were injected through a small abdominal incision into the pancreas head. The needle was removed 10 seconds after completion of the injection, and the incision was closed with absorbable sutures.
--

Other technical points be to be considered are:

1) Concerning the IHC staining, positive control and negative control are needed to exclude false negative and false positive.

Answer:

Positive and negative control are added for IHC staining (**Supplemental Data Fig.S15**). Murine small intestines were used for positive staining of α SMA, CD31, and Ki67. Staining without primary antibody was used as negative control and subjected to DAB visualization. Murine pancreatic tumor derived from KPC transgenic model was used for positive staining of HABP1 and FAP α . KPC tumors were known to be positive for hyaluronic acid (stained by HABP1, *Cancer Cell. 2012 Mar 20; 21(3): 418–429*) and FAP α (*PNAS. 2013. 110(50): 20212–20217*).

Fig. S15. Positive and negative controls for IHC staining.

2) In figure 4C, in the result of WB of CA-IX, lanes in the control, as well as in the IRE + 6Days group, seems to be in the cutting margin of members. This may lead to miscalculation of staining area. And the lanes in the WB of CA-IX are not located in one line. Additionally, the quantification and statistical analyses of WB results in figure 4C may demonstrate the difference between groups in a more precise manner.

Answer:

The western blot of CA-IX was repeated. Statistical analyses are added next to the western blot images in **New Fig. 6D**.

New Fig. 6D. Immunoblotting of tumor lysates (n=3 per group) for hypoxia-inducible factor 1 alpha (HIF-1α), carbonic anhydrase 9 (CA-IX), αSMA, hyaluronic acid binding protein 1 (HABP1), lysyl oxidase (LOX), PDL1, and beta actin (β-actin), and corresponding quantification.

3) On page 7 line 157, there appear to be two pitfalls in the comparison of DC activation markers between IRE-treated group and PBS group. First, the better control group against IRE-treated cells may be cells which did not be treated by IRE. Second, in the legend of figure 3C, does the "CCD7" mean CCR7?

Answer:

Live KRAS* cells are used as control in the **New Fig. 5C**. The typo "CCD7" was corrected to "CCR7".

New Fig. 5C. Representative expression of DC activation markers, including CD40, MHC-II, CCR7, and CD86, on bone marrow-derived DCs (CD11c⁺) after incubation with live or IRE-treated KRAS* cells for 24 hours. Expression was quantified via geometry mean fluorescence intensity (MFI; n = 3). Data are presented as mean ± standard error of mean (SEM). Significance was determined by using the Student *t*-test. **p* < 0.05, ***p* < 0.01, *****p* < 0.0001.

4) In the figure 6E, the cell number of CD4⁺ T cells in control group is similar with that in the IRE+anti-PD-1 group. However, in the figure 7C, the percentage of CD4E (although statistically insignificant) and CD4M in treated group is higher than that in control group. Is there any difference in the measuring methods? Otherwise, what is the cause for this discrepancy? Please clarify this.

Answer:

We apologize for the confusion. The CD4⁺ T cells of previous **Fig. 6 (New Fig. 3E)** are intratumoral cells isolated from KRAS* tumor at 9 days after initiation of treatment. The CD4E/CD4M T cells in previous **Fig. 7 (now New Fig. 4C)** are collected from spleen at 4 months after the initiation of treatment in order to investigate the systemic memory effects. Therefore, the two results are not comparable.

5) In the method section on page 23 line 430, PD-1 or CTLA-4 antibody is administrated intraperitoneally, which is different from the way in clinic and the instruction of clinically used PD-1 antibody product. And this may create gap in the translation of the promising results in this study. Why not choose intravenous injection, for example, tail vein injection?

Answer:

The reviewer is correct that the antibodies are intravenously infused in clinics. However, we would like to point out that the intraperitoneal (i.p.) injection of immune checkpoint antibodies (anti-PD1, anti-CTLA4, etc.) is also commonly used in preclinical studies (*Cancer Cell. 2014 Jun 16; 25(6): 719–734; Nat Med. 2016 Aug; 22(8): 851–860; etc.*). In addition, tail-vein injection for multiple times may damage the blood vessel and cause clogs that can reduce the actual amount of injected antibodies.

6) On page 26 line 273, when demonstrating the long term T cells, the author used treatment-naive healthy mice as control. The better control group here should be that age-matched healthy mice went through sham surgery in the same time before tumor cell re-challenge as treated group.

Answer:

We apologize for the confusion. The reviewer is correct that all control mice were age-matched, healthy mice that went through sham surgery at the same time of the treated mice. The information is updated in the revised manuscript.

7) The figure 1A seems not have been cited throughout the manuscript.

Answer: Fig. 1A is cited as the following: "the experiment set up and treatment schedules are illustrated in **Fig. 1A.**"

8) On page 19 line 340, excessive deposition of hyaluronic acid in PDAC stroma elevates interstitial fluid pressure, which then compresses blood vessels to restrict intratumoral delivery of drugs or immune cells. A citation is needed to support this idea, as it cannot be interpreted from results.

Answer:

A Reference is added. Jacobetz, M. A. et al. Hyaluronan impairs vascular function and drug delivery in a mouse model of pancreatic cancer. *Gut 2013, 62, 112-120.*

Comments from Reviewer #3

The manuscript by Zhao et al assesses the impact of irreversible electroporation on immune checkpoint blockade in animal models. The trust of the paper is that irreversible electroporation induces immunogenic cell death and changes the TME to facilitate response to PD1 checkpoint therapy. While a very interesting observation with significant efficacy observed in the animal models run, I find several aspects of the paper that diminish my enthusiasm in the current form.

Answer:

We thank the reviewer for the positive evaluation. Additional experiments were performed per reviewer's suggestions. We appreciate the opportunity to further improve our manuscript.

Major points.

(1) The largest aspect of this paper that is troubling, is it is mostly observational in nature and not mechanistic. I find no single set of data that firmly explains why IRE improves response to PD1. There is some data for immunogenic cell death (in-vitro) and some data for TME modulation, but neither are formally linked (with data) to the improvement in response. Thus, these things happen, but what is their impact or IRE that unlocks immunotherapy? This significantly impacts my enthusiasm.

Answer:

The manuscript has been re-organized and added with more mechanistic studies. In the **new Fig.3**, we first demonstrated that the anti-tumor efficacy of IRE + anti-PD1 was associated with an increase in the tumor infiltration by CD8⁺ T cells (**new Fig. 3B**), proliferating CD8⁺ T cells (**new Fig. 3C**), and higher CD8-to-Treg ratio (**new Fig. 3D**).

To examine whether the anti-tumor efficacy was a direct result of CD8⁺ T cells, we performed a depletion study to deplete CD8⁺ T cells by injecting a CD8-neutralizing antibody (anti-CD8 α , **new Fig. 3L**). The median survival in the IRE + anti-PD1 + anti-CD8 α group was only 10 days, significantly shorter than the 31.5 days in the IRE + anti-PD1 group ($p < 0.0001$). Therefore, we provide a direct evidence that the CD8⁺ T cells unlocked by the combination of IRE + anti-PD1 was the key contributor to the anti-tumor efficacy.

The depletion results also linked the efficacy data to the mechanistic studies in the latter part of the manuscript. We explored two possible reasons why T cell infiltration was improved by IRE + anti-PD1. We first demonstrated that IRE-treated cells were immunogenic and activated dendritic cells (**new Fig. 5**). Then we showed that IRE modulated the stroma of PDAC that was favorable for T cell infiltration (**new Figs. 6 and 7**).

In addition, we compared IRE with CT-guided X-ray irradiation (10Gy) in their ability to enhance anti-PD1 therapy in KRAS* tumors (**new Figs. 8 and 9**). The results suggested that IRE + anti-PD1 was superior to radiation (10 Gy) + anti-PD1 by providing long-term animal survival (**new Fig. 8E, new Fig. 4D**). The two regimens were also different with regard to mode of cell death and stroma modulation (**new Fig. 9**).

New Fig. 3. Profiling of intratumoral immune cells in KRAS* tumors treated with IRE and/or anti-PD1.

Mice were assigned to sham control, anti-PD1, IRE, or IRE + anti-PD1 groups. (A) Tumor weights. (B) Frequency of CD8⁺ T cells (CD3⁺CD8⁺). (C) Frequency of proliferating CD8⁺ T cells (CD8⁺Ki67⁺). (D) CD8⁺-to-Treg cell ratio. (E) Frequency of CD4⁺ T cells (CD3⁺CD4⁺). (F) Frequency of Tregs (CD4⁺CD25⁺Foxp3⁺). (G) Frequency of NK cells (NK1.1⁺). (H) Frequency of B cells (CD19⁺). (I) Frequency of DCs (CD11c⁺CD11b⁻). (J) Frequency of MDSCs (CD11c⁺CD11b⁺Ly6C⁺ or CD11c⁺CD11b⁺Ly6G⁺). (K) Frequency of macrophages (F4/80⁺). Five tumors per group from the sham control, anti-PD1, and IRE groups, and 6 tumors from IRE + anti-PD1 group were collected 9 days after initiation of treatments. Data are presented as mean \pm standard error of mean (SEM). Significance of differences was determined using 1-way ANOVA followed by Tukey post hoc analysis. (L) Kaplan-Meier survival curves showing the effect of CD8 neutralization. Mice in the IRE + anti-PD1 + anti-CD8 α (n = 7) had significantly shorter survival compared to that in the IRE + anti-PD1 groups (n = 11), log-rank test. *p < 0.05, **p < 0.01, ***p < 0.001, ****p < 0.0001, n.s. not significant.

(2). Is IRE superior or equivalent to other mediators of cell death, like chemotherapy or radiation in its ability to improve PD1 response. How does it compare, and what aspects of cell death, T cell biology and TME modulators are similar and different?

Answer:

The reviewer is correct that both chemotherapy and radiotherapy are first-line treatments for PDAC, and both of them are being evaluated in combination with immunotherapy. For example, low dose of gemcitabine deplete Tregs, and may enhance immunotherapy (*International Journal of Cancer* 2013, 133, 98-107). In a recent publication, we demonstrated that nanomedicine-mediated stromal modulation could enhance the infiltration of PDAC tumor by CD8⁺ T cells (*ACS Nano* 2018, 12, 10, 9881-9893). We performed an *in vitro* treatment of KRAS*

cells with gemcitabine, a first-line chemotherapy drug for PDAC. Significant cell apoptosis/necrosis was not observed until after 48 hours of incubation (**Supplemental Data Fig. S12**). Therefore, chemotherapy-induced death of KRAS* was much slower than that by IRE, which occurred within 30 minutes of treatment. In addition, chemotherapy commonly suppresses the overall immune system. Therefore, it is not a fair comparison with the local IRE treatment.

Per reviewer's suggestion, we investigated the anti-tumor efficacy in KRAS* tumor model by the combination of radiotherapy and anti-PD1. The results are summarized in the **new Figs 8 and 9**. We first studied *in vitro* the mode of cell death. Cells were analyzed within 30 minutes of radiation (10 Gy) or IRE. The results showed that 10Gy radiation did not induce significant disruption of cell membrane (**New Fig. 8A**) or releasing of intracellular DAMPs (**New Figs. 8B-C**) at the 30-minute time interval. The radiation-treated KRAS* cells were also less effective in activating dendritic cells than the IRE-treated cells (**New Fig. 8D**).

We then performed an *in vivo* study. Radiation dose was focused to the left abdominal space using a CT-guided X-ray irradiator, and a total dose of 10 Gy was delivered. We did not further escalate the radiation dose because of the vital organs surrounding the orthotopic tumors in the pancreas, which limit the delivery of radiation at higher doses. Anti-PD1 was injected following the same schedule as that in the IRE + anti-PD1 regimen.

New Fig. 8E shows that radiation + anti-PD1 led to a similar median survival with IRE + anti-PD1, but did not provide benefit of long-term survival. In contrast, IRE + anti-PD1 led to a long-term survival of KRAS*-bearing mice. Out of the 10 re-challenged long-term surviving mice, 6 mice were sacrificed to study the responses in memory T cells at 4 months after the initiation of treatment. The remaining 4 mice were monitored for extended survival, and all of them survived for more than 9 months from the initiation of treatment. Histological examination found no microscopic tumor nodules in their pancreas (**Fig. 4D, Supplemental Data Fig. S10**).

Analyses of tumors at 9 days after treatment initiation showed that radiation + anti-PD1-treated tumors were larger than those treated by IRE + anti-PD1 (**New Fig. 8F**), and had fewer infiltrating CD8⁺ T cells (**New Fig. 8G**). Tumor sections revealed that IRE + anti-PD1 induced substantial necrosis (**New Fig. 8J**), while radiation + anti-PD1 did not change the stromal components as compared to sham control (**New Fig. 9**).

In conclusion, we proved that IRE + anti-PD1 was superior to radiation + anti-D1 by leading to long-term survival.

Figure 8. Comparison between radiation and IRE in the enhancement of anti-PD1 therapy in KRAS* tumor. (A) Annexin V-FITC/PI staining of KRAS* suspended in phosphate buffered saline (PBS) after 10 Gy of radiation, or electroporated using 960-V, 100- μsec electric pulses at 1 Hz frequency for 20 pulses in a 4 mm-gap cuvette. (B&C) ATP and HMGB1 concentrations in the supernatants of electroporated cells ($n = 3$). Cells were analyzed within 30 minutes of treatments (D) Representative expression of DC activation markers, including CD40, MHC-II, CCR7, and CD86, on bone marrow-derived DCs (CD11c⁺) after incubation with KRAS* cells for 24 hours. Expression was quantified via geometry mean fluorescence intensity (MFI; $n = 3$). (E) Kaplan-Meier survival curves of mice in sham control ($n = 8$), 10 Gy + anti-PD1 ($n = 11$), and IRE + anti-PD1 groups ($n = 11$). (F-I) Tumor weight, frequency of intratumoral CD8⁺ T cells and Treg, and CD8-to-Treg ratios from 4 to 6 tumors per group collected at 9 days after radiation or IRE. (J) Representative hematoxylin-eosin staining of tumor sections and quantification of tumor necrosis. Data are presented as mean \pm standard error of mean (SEM). Significance was determined using 1-way ANOVA followed by Tukey post hoc analysis or Student t -test. * $p < 0.05$, ** $p < 0.01$, **** $p < 0.0001$, n.s. not significant.

Fig. 9. IHC staining of viable tumor region at 9 days initiation of 10 Gy + anti-PD1 or IRE + anti-PD1. Representative micrographs of staining for CD31 (A), Picrosirius Red (B), α SMA (C), FAP α (D), HABP1 (E), and Ki67 (F) and corresponding quantifications. Ten to fifteen 200 \times visual fields were randomly captured. Scale bars = 50 μ m. Five doses of anti-PD1 had been administered at this time point for the 10 Gy + anti-PD1 and IRE + anti-PD1 groups. Data are presented as mean \pm SEM. Significance was determined using 1-way ANOVA followed by Tukey post hoc analysis. * $p < 0.05$, ** $p < 0.01$, *** $p < 0.001$, **** $p < 0.0001$. MFI, mean fluorescence intensity; FOV, field of view; n.s., not significant.

(3). There are aspects of the data presented that should be improved.

A. The use of western blots, as in Figure 4c to show changes in the TME like FAP+ CAFs is not the best method, as it does not take into account cell diversity. Indeed the authors use IHC in Figure 5 and should in the time course analysis in Figure 4.

B. MFI analysis is not likely appropriate for analysis of FAP α CAF staining shown in Figure 5D. Most investigator show area or cell number (this may just be typo).

Answer to A and B:

FAP α IHC staining is added to now **New Fig. 6C** to show changes in TME after IRE treatment. The quantification of FAP α is changed to percentage of FAP α + pixels per visual field throughout the manuscript (**New Figs.6C, 7D, and 9D**).

C. Did the investigators count both viable and non-viable areas in quantitation in Figure 5. And should they not also show data excluding the necrotic area.

Answer:

We apologize for the confusion. All IHCs images throughout the manuscript (now **New Figs. 6, 7, and 9**) are taken from the viable tumor regions. This is now specifically stated in the legend of each figure.

D. The flow cytometry data in figure 3a is not properly compensated. The diagonal staining profile is indicative of this and this aspect would affect the data in all panels. Also was this done in replicates with graphical or merely one time one well/condition. Outcome is believable, but data should be improved.

Answer:

The flow cytometry in Fig. 3A (**now New Fig. 5A**) was repeated with corrected compensation. All representative results are based on multiple independent experiments. The exact number of replicates is added to the legend of each figure panel.

E. The number of repeat experiments done for each figure panel is not noted. Example “representative of 3 independent in vitro experiments”. Thus it’s unclear if the data represent a single outcome or robust repeatable findings. Likely the later but this should be spelled out.

Answer:

We apologize for the confusion. Representative data were based on multiple independent experiments. The exact number of replicates is added to the legend of each figure panel.

4. Why does PD1 work and CTLA4 not work? Is PDL1 more relevant to IRE and why? Or is CTLA4 not relevant to this mouse model (possible)

Answer:

Our results did not exclude the possibility that anti-CTLA4 therapy might also be enhanced by IRE. We chose anti-PD1 because in clinical studies anti-PD1 showed less toxicity than anti-CTLA4 (*Cancer Treatment Reviews 2016, 44, 51-60*). Avoiding excessive toxicity is important in clinical practice, since patients treated with IRE-alone already require extensive post-surgery care (*Med Oncol 2017, 34, 38*). This is the exact reason we did not use the dual-antibody regimen, i.e. anti-PD1 + anti-CTLA4, after IRE. As shown in **Fig. 2**, although the IRE + dual antibody provided marginal increase in survival percentage, the initial drop in body weight (at 5 days after treatment started) was close to 15%.

We did not include anti-PDL1 in our study because our goal was to use anti-PD1 to activate T cells after IRE. In addition, **New Fig. 6D** shows a significant reduction in PD-L1 expression at 4 days after IRE. Therefore, adding anti-PDL1 antibodies is unlikely to further improve the therapeutic efficacy.

Reviewers' comments:

Reviewer #1 (Remarks to the Author):

The authors have adequately responded to all my questions and comments and have modified adequately their manuscript so I have no further comments or questions.

Reviewer #2 (Remarks to the Author):

This reviewer feels satisfied by the thorough, point-to-point revisions made by the authors according to the comments from the reviewer(s).

Reviewer #3 (Remarks to the Author):

Overall the manuscript is noticeably improved. The authors have provided responses with data for all of my queries.

I still feel the data are observational and not mechanisms proving for why IRE improves PD1.

Example 1, the authors show PD1 efficacy is CD8 mediated, does not tell me why the combo derived increase CTL infiltration.

Example 2, the authors show IRE killing in-vitro is more immune stimulatory, but not that any of these pathways matter in-vivo for response to PD1.

That said (and stepping off my soap-box), the authors have put together a satisfying set of observations, that I think are novel and meritorious even without the precise mechanism in hand. I am satisfied.

I only have one more technical concern that I feel should be addressed.

1. Figure 8E has a Kaplan Myer Analysis, it does not look like there is a statistically significant difference between the IRE+PD1 and RT+PD1 survival groups. The legend does not provide a statistical

test or value. The p-value and test should be provided directly for the difference between these two groups (IRE+PD1 and RT+PD1). Possibly showing the survival analysis to day 70 or 75, if data are available, would help emphasize the difference. Since the authors make this to be a significant finding the differences and statistical analysis should be epically clear.

Point-to-Point Responses

Editor's Comments

Your revised manuscript entitled "Irreversible Electroporation Reverses Resistance to Immune Checkpoint Blockade in Pancreatic Cancer" has now been seen by 3 referees. You will see from their comments below that while some important points are raised by the original Reviewer # 3. We are interested in the possibility of publishing your study in Nature Communications, but would like to consider your response to these concerns in the form of a revised manuscript before we make a final decision on publication.

We therefore invite you to revise and resubmit your manuscript, taking into account the points raised. Please highlight all changes in the manuscript text file. In particular, we do not require that you perform additional experiment to further investigate the mechanism underlying the improvement of PD1 by IRE. However, please address the technical concern regarding the statistical analysis used in Figure 8E.

Answer:

The manuscript is revised per editor's and reviewers' suggestions. Point-to-point responses are provided below.

(1) To improve the quality of methods and statistics reporting in our papers, we are now asking all authors to complete an editorial policy checklist that verifies compliance with all required editorial policies. Please ensure that the checklist is completed and uploaded with your revised article. All points on the policy checklist must be addressed; if needed, please revise your manuscript in response to these points. Please note that this form is a dynamic 'smart pdf' and must therefore be downloaded and completed in Adobe Reader, instead of opening it in a web browser.

Editorial policy checklist: <https://www.nature.com/authors/policies/Policy.pdf>

Answer:

All authors have signed the policy checklist. The combined PDF file is provided as attachment.

(2) At the same time, we ask that you ensure your manuscript complies with our editorial policies. Please ensure that the following requirements are met, and any relevant checklist is completed or updated and uploaded as a Related Manuscript file type with the revised article.

Reporting requirements for life sciences research:

<https://www.nature.com/authors/policies/ReportingSummary.pdf>

Answer:

The reporting summary is provided as attachment. The manuscript is revised to comply with editorial policies. Specifically: (1) Scattered plots are used in all possible cases, e.g. **Figure 3**. (2) Individual data points are overlaid with bar graphs, e.g. **Figure 4B**. (3) When individual data points cannot be overlaid due to limited spaces, the data is presented as interquartile range (IQR) with a median center line and min to max error bars to show the data distribution, e.g. **Figures 2C** and **4A**. (4) Gating strategy for flow cytometry is provided in the reporting summary. Representative flow cytometry pseudo-colored plots are provided as examples in the supplemental materials (**Figures S7, 8, 9 & 13**).

(3) In an effort to ensure reproducibility of research data, we now also require that you provide a separate source data file. The source data file should, as a minimum, contain the raw data underlying all reported averages in graphs and charts, and uncropped versions of any gels or blots presented in the figures. To learn more about our motivation behind this policy, please see <https://www.nature.com/articles/s41467-018-06012-8>.

Within the source data file, each figure or table (in the main manuscript and in the Supplementary Information) containing relevant data should be represented by a single sheet in an Excel document, or a single .txt file or other file type in a zipped folder. Blot and gel images should be pasted in and labelled with the relevant panel and identifying information such as the antibody used. We also encourage you to include any other types of raw data that may be appropriate. An example source data file is available demonstrating the correct format: <https://www.nature.com/documents/ncomms-example-source-data.xlsx>

The file should be labeled 'Source Data', with the title and a brief description included in your cover letter, and should be mentioned in all relevant figure legends using the template text below.

"Source data are provided as a Source Data file."

Answer:

The raw data underlying all reported averages are included in the Excel file titled "Source Data". The original data for each figure are presented in a single Excel sheet. The original western blot images with

antibody information are pasted into a separate sheet of the Excel file. They are also included in **Supplementary Fig. S16** per the requirement by format checklist.

(4) Furthermore, your manuscript should comply with our format requirements, which are summarized on the following checklist: <https://www.nature.com/documents/ncomms-manuscript-checklist.pdf>

Answer:

The manuscript is re-formatted to comply with the checklist. The format checklist is completed and provided in the attachment.

(5) We noticed that your paper appears to rely on the use of custom code/software. We would like to clarify if and how the software/algorithms necessary to reproduce the results will be made available to the scientific community upon publication as required by our material sharing requirements. For more information on this please see <http://www.nature.com/authors/policies/availability.html#code>

In order for the reviewers to evaluate the work adequately they must be able to test the software/review the code themselves. If you have not yet provided the software, we therefore request that you provide a single compressed zip file containing the software with a readme.txt file or other user manual containing complete instructions for installing and running the software. If appropriate, please also provide example data and expected output. Sufficient material should be provided for referees to directly test the performance of the software/algorithm.

If the software and materials are small enough to fit in a single compressed zip file less than 6MB in size, you may email this file directly to me. If the zip file is between 6 MB and 200 MB you may upload it to our file transfer site. If necessary, a second zip file up to 200 MB in size can be used to supply the example data. Please let me know if you need to use this option and I'll send you further details.

Please also fill out and return to me the code and software submission checklist that will be made available to editors and reviewers during manuscript assessment. Please note that this form is a dynamic 'smart pdf' and must therefore be downloaded and completed in Adobe Reader, instead of opening it in a web browser. <https://www.nature.com/authors/policies/Software.pdf>

Answer:

We apologize for the confusion. However, our study did not involve any custom code or software. All software used in this study and corresponding license information are summarized in the table below:

Software Name and License Information	Application
GraphPad Prism 7. Institutional license	Animal survival, dot plots, bar graphs, and associated statistical analyses (Figs. 1, 3, etc.). Software usage follows the manufacturer's instructions
Image Scope v12.3.2.8013 Downloaded from http://www2.leicabiosystems.com/l/48532/2014-11-18/35cqc	Viewing scanned H&E tissue sections (Fig. 8J) with scale bar information
Bruker Biospin software. Associated with the Biospec USR70/30 MRI system	Acquiring T ₂ -MRI images (Fig. 1D).
Odyssey Infrared Imaging System Application Software. Commercial software associated with the Odyssey near-infrared fluorescence scanner (LI-COR).	Acquiring western blot images (Fig. 6D). Fluorescent secondary antibodies of 700 nm or 800 nm channels were used for western blot. Excitation voltage, brightness, and contrast were adjusted using the software per manufacturer's instructions.
Image Reader FLA-5000 series version 1.0. Commercial software associated with the FLA-5000 image reader.	Fluorescence scanning of tissue sections injected with FITC-dextran in Fig. 6E . Excitation/emission filters, and excitation voltages were selected using the software per manufacturer's instructions.

Multi Gauge. Commercial software associated with the FLA-5000 image reader.	Quantification of mean fluorescence intensity of tissue sections in Fig. 6E .
Flow Jo V10 , Institutional license	Flow cytometry analysis following manufacturer's instructions. The gating strategies and representative pseudo-colored plots are included in Supplementary Information (Figs. S7-S9, etc.).
AxioVision Microscope Software 4.8. Commercial software associated with the Axio Observer.Z1 fluorescence microscope	Acquiring microscopic imaging (Figs. 7, 9, etc.). Imaging collection parameters, including exposure time, brightness, white balance, and contrast, etc. are adjusted using the software following manufacturer's instructions, and kept identical among different treatment groups
Image J 1.52a , Public domain software downloaded from https://imagej.nih.gov/ij/download.html Color Deconvolution plugin Downloaded from https://imagej.net/Colour_Deconvolution	Analyses of IHC microscopic images (Figs. 7, 9, etc.) per developer's instructions, including measuring the percentage of positive staining pixels and adding scale bars.

Below is an example of measuring the percentage of CD31⁺ pixels in an IHC section using Image J 1.52a with the color deconvolution plugin. The general principle is described at <https://imagej.nih.gov/ij/docs/examples/stained-sections/index.html>.

(6) <DATA AVAILABILITY PARA> Data availability statements and data citations policy: All Nature Communications manuscripts must include a section titled "Data Availability" as a separate section after the Methods section but before the References. For more information on this policy, and a list of examples, please see <http://www.nature.com/authors/policies/data/data-availability-statements-data-citations.pdf>. In particular, the Data availability statement should include:

- Accession codes for deposited data
- Other unique identifiers (such as DOIs and hyperlinks for any other datasets)

- At a minimum, a statement confirming that all relevant data are available from the authors
- If applicable, a statement regarding data available with restrictions
- If a dataset has a Digital Object Identifier (DOI) as its unique identifier, we strongly encourage including this in the Reference list and citing the dataset in the Data Availability Statement.
- If a source data file is provided, please add a reference to this in the data availability statement. For example:
 - “The source data underlying Figs 1a, 2a–d, 6d, h and 7c and Supplementary Figs 1a and 5d are provided as a Source Data file.”

Answer:

A statement of data availability is provided in the manuscript as follows:

All the data supporting the findings of this study are presented within the article and its Supplemental Information files. Data is available from the corresponding author upon reasonable request. The source data underlying Figs 1-9, and Supplementary Figs. S1 and S11, are provided as a Source Data file.

(7) DATA SOURCES: We strongly encourage authors to deposit all new data associated with the paper in a persistent repository where they can be freely and enduringly accessed. We recommend submitting the data to discipline-specific, community-recognized repositories, where possible and a list of recommended repositories is provided here: <http://www.nature.com/sdata/policies/repositories>

If a community resource is unavailable, data can be submitted to generalist repositories such as figshare (<https://figshare.com/>) or Dryad Digital Repository (<http://datadryad.org/>). Please provide a unique identifier for the data (for example a DOI or a permanent URL) in the data availability statement, if possible. If the repository does not provide identifiers, we encourage authors to supply the search terms that will return the data. For data that have been obtained from publicly available sources, please provide a URL and the specific data product name in the data availability statement. Data with a DOI should be further cited in the methods reference section.

Please refer to our data policies here: <http://www.nature.com/authors/policies/availability.html>

Answer:

Our manuscript does not contain any data that requires mandatory data deposit. All the data supporting the findings of this study are presented within the article and its Supplemental Information files. The source data underlying Figs 1-9, and Supplementary Figs. S1 and S11, are provided as a Source Data file.

Reviewer #3 (Remarks to the Author):

Overall the manuscript is noticeably improved. The authors have provided responses with data for all of my queries.

I still feel the data are observational and not mechanisms proving for why IRE improves PD1.

Example 1, the authors show PD1 efficacy is CD8 mediated, does not tell me why the combo derived increase CTL infiltration.

Example 2, the authors show IRE killing in-vitro is more immune stimulatory, but not that any of these pathways matter in-vivo for response to PD1.

That said (and stepping off my soap-box), the authors have put together a satisfying set of observations, that I think are novel and meritorious even without the precise mechanism in hand. I am satisfied.

Answer:

We thank reviewer for the insightful suggestions. We agree that it is important to further investigate the detailed signaling cascades that are responsible for the anti-tumor efficacy of IRE + anti-PD1. However, such a comprehensive study would require high-throughput approaches such as proteomics and genomic sequencing, which are beyond the scope of the current studies. The reviewer's suggestions have been added to our future study plan.

I only have one more technical concern that I feel should be addressed.

1. Figure 8E has a Kaplan Myer Analysis, it does not look like there is a statically significant different between the IRE+PD1 and RT+PD1 survival groups. The legend does not provide a statistical test or value. The p-value and test should be provided directly for the difference between these two groups (IRE+PD1 and RT+PD1). Possibly showing the survival analysis to day 70 or 75, if data are available, would help emphasize the difference. Since the authors make this to be a significant finding the differences and statistical analysis should be epically clear.

Answer:

We apologize for the confusion. The reviewer is correct that there was no significant difference in terms of median survival between the 10 Gy RT + anti-PD1 group and the IRE + anti-PD1 group (30 vs. 31.5 days, $p = 0.16$; Log-rank test). However, the IRE + anti-PD1 treatment did yield long-term survivors (4 out of 11 mice, 36%) that rejected re-challenge with KRAS* cells. These mice were monitored for 120 days (i.e. 60 days of initial survival and 60 days of tumor re-challenge) before further analyses (**Figs. 4A-C**). Therefore, we re-plotted the survival curve for the IRE + anti-PD1 treatment group in **Fig. 1A** up to 120 days together with that of the 10 Gy + anti-PD1 group into the **New Fig. 8E**.

New Fig. 8E. Kaplan-Meier survival curves of mice in sham control (n = 8), 10 Gy + anti-PD1 (n = 11), and IRE + anti-PD1 groups (n = 11). There was no significant difference in terms of median survival between 10 Gy + anti-PD1 and IRE + anti-PD1 ($p = 0.16$, log-rank test).

We conducted additional MRI study on 5 mice treated with IRE + anti-PD1, which were enrolled on 09/18/18 (**Fig. 1D, Supplementary Figs. S6**). Two of the 5 mice (40%) have survived for 75 days by 12/02/18 without palpable tumors. These newer data further support that combined IRE + anti-PD1 therapy produces durable response in PDAC.

We enrolled in a total of 10 long-term surviving mice for the re-challenge studies (**Fig. 4A**). Six of the ten mice were euthanized to study the responses in memory T cells at 4 months after the initiation of treatments (**Figs. 4B-C**). The remaining 4 mice were monitored for extended survival. All of them survived for more than 9 months from the initiation of treatment. Histological examination revealed no microscopic tumor nodules in their pancreas (**Fig. 4D, Supplementary Fig. S10**). Therefore, we conclude that IRE + anti-PD1 induced a curative response against PDAC.

REVIEWERS' COMMENTS:

Reviewer #3 (Remarks to the Author):

My single remaining query was answered well. Sorry for the delay.